# Huxley-Gödel Machine: Human-Level Coding Agent Development by an Approximation of the Optimal Self-Improving Machine

**Wenyi Wang**[*]  **Piotr Piękos**[*]  **Li Nanbo**  **Firas Laakom**  **Yimeng Chen**
**Mateusz Ostaszewski**  **Mingchen Zhuge**  **Jürgen Schmidhuber**

{wenyi.wang, piotr.piekos, nanbo.li, firas.laakom, yimeng.chen,
mateusz.ostaszewski, mingchen.zhuge, juergen.schmidhuber}@kaust.edu.sa
King Abdullah University of Science and Technology (KAUST)
Thuwal, Saudi Arabia

## Abstract

Recent studies operationalize self-improvement through coding agents that edit their own codebases. They grow a tree of self-modifications through expansion strategies that favor higher software engineering benchmark performance, assuming that this implies more promising subsequent self-modifications. However, we identify a mismatch between the agent's self-improvement potential (metaproductivity) and its coding benchmark performance, namely the *Metaproductivity-Performance Mismatch*. Inspired by Huxley's concept of clade, we propose a metric (CMP) that aggregates the benchmark performances of the *descendants* of an agent as an indicator of its potential for self-improvement. We show that, in our self-improving coding agent development setting, access to the true CMP is sufficient to simulate how the Gödel Machine would behave under certain assumptions. We introduce the Huxley-Gödel Machine (HGM), which, by estimating CMP and using it as guidance, searches the tree of self-modifications. On SWE-bench Verified and Polyglot, HGM outperforms prior self-improving coding agent development methods while using fewer allocated CPU hours. Last but not least, HGM demonstrates strong transfer to other coding datasets and LLMs. The agent optimized by HGM on SWE-bench Verified with GPT-5 mini and evaluated on SWE-bench Lite with GPT-5 **achieves human-level performance, matching the best officially checked results of human-engineered coding agents**. Our code is publicly available at https://github.com/metauto-ai/HGM.

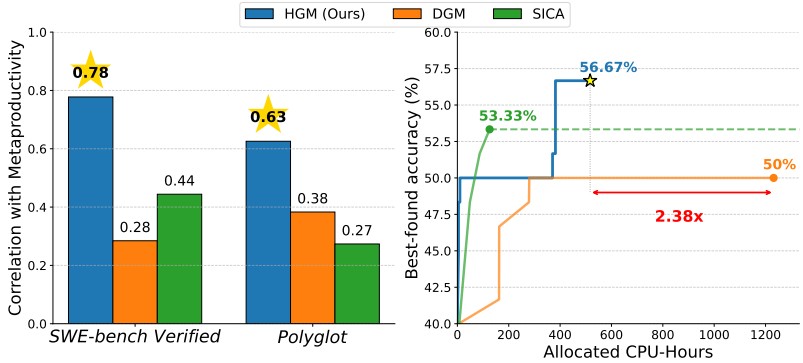

Figure 1: **(Left)** Weak correlation between the guidance metrics of other methods (based on performance) and long-term self improvement; HGM mitigates this mismatch by leveraging clade-level metaproductivity. **(Right)** On *SWE-bench Verified*, HGM achieves **higher** accuracy with 2.38 times less allocated CPU-hours. SICA encountered repeated errors after consuming 45% of its budget.

---

[*]equal contribution

# 1 INTRODUCTION

Processes of self-modification drive the growth of complex systems, from biological evolution (Hendrikse et al., 2007; Dawkins, 2019) to cultural and scientific innovation (Good, 1966; Hall, 2007). These general ideas have been instantiated in concrete algorithms for self-improving agents (Schmidhuber, 1987; 2003; Nivel et al., 2013; Everitt et al., 2016), demonstrating how abstract principles of self-modification can be translated into operational mechanisms. Unlike static systems constrained by fixed architectures, such agents can incrementally modify their own self-modification mechanisms and learning strategies, reusing newly gained abilities to fuel subsequent improvements. This capacity fosters continual adaptation, reduces reliance on human intervention, and enables problem-solving capabilities that cannot be fully anticipated at design time.

A central challenge is how to decide which self-modifications to accept. The Gödel machine (Schmidhuber, 2003) (GM) offers a theoretically optimal answer: accept only modifications that provably increase the expected long-term utility. While this provides a sound blueprint, its reliance on formal proofs makes it practically challenging. Recent implementations instead rely on coding agents that edit their own codebases and favor self-modifications from agents with higher benchmark performance (Robeyns et al., 2025; Zhang et al., 2025a). Yet, as illustrated in Figure 1 (left), this heuristic can be misleading: a high-scoring agent may produce unproductive descendants, while a lower-scoring one seeds lineages that achieve greater long-term gains. We term this phenomenon the *Metaproductivity-Performance Mismatch*.

To address this mismatch, we introduce *clade-level metaproductivity* (CMP), inspired by Huxley's notion of clades as lineages of common ancestry (Huxley, 1957). A clade is a group of organisms that includes a common ancestor and all of its descendants; for example, mammals form a clade, and primates form a clade within mammals. CMP quantifies the productivity of a clade by aggregating the success of an agent's descendants rather than relying solely on its immediate benchmark score. Furthermore, we show in Theorem 1 that in our self-improving coding agent development setting (Assumption 1, which includes the assumption that the only quality of the self-improvement process is the evaluation score of the final agent and that the evaluation is conducted with repeatable trials), having access to the true CMP oracle suffices to imitate the Gödel Machine.

This insight motivates our proposed algorithm, the **Huxley-Gödel Machine (HGM)**, which approximates GM-style self-improvement by estimating CMP from clade-aggregated descendant outcomes and selecting nodes to expand via Thompson sampling (Thompson, 1933). Furthermore, by leveraging a more reliable estimate, we adaptively decouple expansion from evaluation, leading to asynchronous execution for efficient parallelism.

Empirically, HGM better aligns with long-run agent productivity than benchmark-driven baselines, as shown in Figure 1 (left). On SWE-bench Verified (Jimenez et al., 2024; Chowdhury et al., 2024) and Polyglot (Gauthier, 2024), HGM consistently outperforms Darwin Gödel Machine (DGM) (Zhang et al., 2025a) and Self-Improving Coding Agent (SICA) (Robeyns et al., 2025). Remarkably, one agent found by HGM surpasses SWE-agent (Yang et al., 2024), the highest-scoring human-engineered coding agent with officially checked results, on SWE-bench Lite (Jimenez et al., 2024), when both use the GPT-5 mini backbone under matched budgets. The HGM-discovered agent transfers robustly when evaluated under a shift that is simultaneous in both the dataset and the model. Although optimized on SWE-bench Verified with GPT-5 mini, when tested on SWE-bench Lite with the GPT-5 backbone, **it achieves performance on par with the best officially verified human-engineered coding agents**.

To summarize, our contributions are as follows:

- We analytically define the Clade-Metaproductivity (CMP) function as a measure of agents' self-improving ability and show that in a self-improving coding agent development setting (Assumption 1), access to a CMP oracle suffices to reproduce the Gödel Machine's acceptance mechanism. (Theorem 1).

- We empirically observe that immediate benchmark performance is an unreliable predictor of CMP and show that our CMP estimator aligns better.

- Using our CMP estimator, we propose the **Huxley-Gödel Machine (HGM)**, which approximates the Gödel Machine in a coding agent setting from partial evaluations and guides the expansion via Thompson sampling with adaptive scheduling.

- We empirically validate HGM on SWE-bench Verified and Polyglot, demonstrating higher-quality optimized agents compared to previous self-improving methods, even though they were discovered within substantially smaller allocated CPU-hours. Furthermore, HGM **achieves human-level coding agent design** on SWE-bench Lite by optimizing on SWE-bench Verified.

## 2 SELF-IMPROVING AS TREE-SEARCH

Both the Darwin Gödel Machine (DGM) and the Self-Improving Coding Agent (SICA) belong to the class of self-referential AI (Schmidhuber, 1987; 2006), wherein a single agent modifies itself to generate new agents, each empirically validated on downstream tasks. In this paper, we formalize this self-improvement process as an iterative tree-search problem, where the goal is to discover an agent that maximizes performance across multiple downstream tasks. Concretely, starting from an initial agent as the root, a tree-search policy incrementally grows the tree of self-modified agents. At each iteration, the policy either selects an agent (a node in the tree) to expand by producing a child agent (a self-modified version of the selected agent) or selects an agent to undergo additional evaluation on downstream tasks.

Formally, let $\mathcal{T}_t$ denote the archive of our agents at iteration $t$. In this paper, the archive is always represented as a tree of evolved agents, and we use the terms archives and trees interchangeably. $T_0 = \{a_0\}$ is initialized as a single-node tree with a fixed initial agent. At iteration $t$, the policy selects actions $a_{t+1} \sim \pi(\cdot \mid \mathcal{T}_t)$, where $\pi$ is a policy over actions $\mathcal{A}_t = \mathcal{M}_t \cup V_t$, where $\mathcal{M}_t = \{m_a : a \in \mathcal{T}_t\}$ are agent modifications and $V_t = \{v_a : a \in \mathcal{T}_t\}$ are evaluations. Here, $m_a$ instructs agent $a$ to produce a self-modification that is added as a child to the tree, while $v_a$ selects an agent from the tree for an additional evaluation on one more downstream task. After exhausting the computational budget, the policy selects a final agent ($a_{\text{final}} = \arg\max_{a \in \mathcal{T}} Score_\pi(a) \in \mathcal{T}_T$ where $T$ is the termination iteration and $Score$ is part of the policy) from the final tree as the returned agent. The objective is to optimize $J(\pi) = \mathbb{E}[U(a_{\text{final}})]$, where $U$ is a utility function that measures downstream tasks performance. In this work, we define $U$ as the average of binary success indicators across all downstream tasks. $\pi$ denotes an algorithm, with DGM, SICA, and our proposed HGM representing concrete instances.

**Compound Policy.** At each step of self-improvement, the system faces a compound decision: whether to expand the tree by generating new agents or to evaluate existing ones. This decision naturally decomposes into three sub-policies: (i) a *selection policy* that chooses between expansion and evaluation, (ii) an *expansion policy* that determines which parent to modify, and (iii) an *evaluation policy* that selects which agent to test. Prior approaches, such as SICA and DGM, conflate these choices. They always expand a parent, create a child, and immediately evaluate that child on multiple tasks. This fixed sequence restricts flexibility: once a new agent is generated, it monopolizes evaluations, even if older agents appear more promising. For instance, an agent that fails nine tasks in a row continues to consume evaluations, while an older agent with partial successes is ignored.

HGM breaks this rigidity by *decoupling* expansion from evaluation. At each step, it adaptively decides whether to generate a new agent or to further probe an existing one, and evaluations are always at the granularity of a *single agent-task pair*. This finer control enables early stopping on unpromising agents. Table 4 summarizes how SICA, DGM, and HGM instantiate these sub-policies.

## 3 HUXLEY-GÖDEL MACHINE

The original Gödel Machine is a general task solver that can, in principle, optimally perform any provable self-improvements in a computable environment with respect to a given objective (Schmidhuber, 2003). It operates by running a proof searcher that continually seeks formal proofs that modifying its own code will increase expected utility; once such a proof is found, the modification is executed and permanently changes the machine (or program). Crucially, the theoretical analysis accounts for the single-life setting (no repeatable trials) and the real cost of proof search in time and resources that could otherwise be spent collecting reward. In contrast, this paper considers a setting tailored to self-improving coding agent development under Assumption 1, where the objective is solely the utility of the final agent selected at the end of development, evaluations are conducted in a repeatable test environment reset between trials, each self-modification reduces the remaining

time budget by exactly one unit, and the only operation incurring time cost is the self-modification itself. Within this framework, the Gödel Machine can be viewed as an optimal agent operating in a POMDP in which the policy observes only the parent $a_{\text{parent}}$, the child $a_{\text{child}}$, and the remaining budget $b$, chooses whether to accept or reject the child, and at termination $Score_\pi$ selects either the final parent or child as output; a full POMDP specification is provided in Appendix A.

> **Assumption 1.** *For the theoretical analysis of Gödel Machine applied to self-improving coding agents, we make the following additional assumptions in comparison to the setup from the original Gödel Machine:*
>
> - *The policy objective function is defined as a function of only the final agent, with no other rewards received before termination;*
> - *The agent's utility is measured by its performance on evaluation tasks, under the assumption of repeatable trials: for any agent-task pair, the expected outcome is independent of evaluation time or prior events.*
> - *The proofs of Gödel Machines do not consume budget;*
> - *And each self-modification costs exactly one unit of the budget.*

Since formal proof search of self-improvements can be intractable, in this section, we introduce the Huxley-Gödel Machine (HGM), a self-improving machine that approximates Gödel Machine by using clade-level statistics. At the core of HGM is metaproductivity, a measure of agents' ability to improve their self-improvement skills that lead to downstream performance of distant future agents.

In Section 3.1, we introduce two metrics of metaproductivity: Global metaproductivity (GMP), which captures how evolving a given agent increases the metaproductivity of the entire tree of agents. This measure of metaproductivity is general and hard to operationalize or estimate. We instead introduce clade-metaproductivity (CMP) that measures only how promising evolutions starting from a given agent (its clade) are. In Theorem 1, we show that access to true CMP is sufficient to implement a Gödel Machine applied to the coding agent development setting (Assumption 1). Following on that, in Section 3.2, we introduce the Huxley-Gödel Machine (HGM), that guides the self-improvement search with Thompson Sampling based on the estimate of CMP.

## 3.1 METAPRODUCTIVITY AND CLADE-METAPRODUCTIVITY

Given a policy $\pi$, to quantify the quality of how an agent's self-modification influences the performance of the system, we define the notion of global metaproductivity (GMP):

$$\text{GMP}_\pi(\mathcal{T}, a) \ = \ \mathbb{E}_{\mathcal{T}_B \sim p_\pi(\cdot|\mathcal{T}, a)} \left[ U(\text{argmax}_{a' \in \mathcal{T}_B} Score_\pi(a')) \right],$$

where $\mathcal{T}$ is a tree of agents and $a \in \mathcal{T}$. $Score_\pi$ is the function that scores the agents for the final selection. The policy $\pi$ unrolls the trajectory until the end of the episode with policy $\pi$ and produces a final archive of agents $\mathcal{T}_B$. The distribution of the trajectory is given by $p_\pi$.

GMP directly corresponds to the Q-value function in reinforcement learning, with state phrased as the archive of agents, and action being the selected agent to expand. The GMP value of a node measures how good (on average) the final agent obtained from the search process will perform. GMP measures the long-term potential of self-improvements, which also includes modifications that improve self-improvement itself and so on. An algorithm might, at the beginning, focus on improving the ability to self-improve while neglecting direct benchmark abilities, only to later focus on them. This is a principal meta-learning behavior that is captured in the original Gödel Machine (Schmidhuber, 2003). The objective of designing a policy for self-improvement (Section 2) is equivalent to optimizing $\text{GMP}(\{a_0\}, a_0)$.

While GMP captures the full long-term potential of a policy, its scope is overly broad for practical conceptualization. Notably, the Gödel Machine considers only the outcomes of the current agent and its descendants when deciding whether to accept a modification. Motivated by this observation, we define a localized variant of GMP that focuses on the subtree rooted at a given agent, i.e., its *clade*. We refer to this quantity as Clade-Metaproductivity (CMP):

$$\text{CMP}_\pi(\mathcal{T}, a) = \mathbb{E}_{\mathcal{T}_B \sim p_\pi(\cdot | \mathcal{T}, a)} \left[ U(\text{argmax}_{a' \in \mathcal{C}(\mathcal{T}_B, a)} Score_\pi(a')) \right]$$

$$= \mathbb{E}_{\mathcal{T}_B \sim p_\pi(\cdot | \mathcal{T}, a)} \left[ \text{max}_{a' \in \mathcal{C}(\mathcal{T}_B, a)} U(a') \right] \qquad \text{(if Sel = U)},$$

where $\mathcal{C}(\mathcal{T}_B, a)$ is the clade (i.e., the subtree with $a$ as the root) of the node $a$ in the Tree $\mathcal{T}_B$ and $Score$ is the final agent selection metric.

CMP contains the non-greedy information about the future evolution of self-improving agents, therefore guiding good strategies for self-improvement aimed also at the improvement of the self-improvement itself. Furthermore, we show the crucial relation of CMP to the Gödel Machine.

> **Theorem 1.** *Under Assumption 1, access to the* CMP *oracle is sufficient to implement the Gödel Machine.*

The proof is available in the App. A. This observation motivates us to introduce the estimate of CMP and use this as guidance in our algorithm. An algorithm with a perfect estimate of CMP would be able to produce the Gödel Machine. HGM by estimating CMP approximates the original Gödel Machine. We describe our algorithm fully in the next section.

### 3.2 ALGORITHM

Existing methods use benchmark performance on coding tasks as a guidance metric, treating task success as an indicator of self-improvement potential. This assumption is overly greedy: it evaluates only the immediate utility of a modification while ignoring its downstream consequences for future self-modifications. We refer to this gap as the *Metaproductivity-Performance Mismatch*: the divergence between short-term task performance and the long-term capacity for self-improvement as measured by CMP. Empirical evidence shows that this mismatch happens in practice (see Section 4.1.) We aim to model long-term, global dependencies by deriving our estimator of CMP. Specifically, we define HGM by stating its three subpolicies.

**Expansion Policy** The core of the HGM algorithm is its selection criterion for expansion. HGM aims to estimate Clade-Metaproductivity with the motivation that the true CMP as the criterion would produce the Gödel-Machine due to Theorem 1. In this sense, HGM approximates Gödel-Machine, the **optimal** self-improving machine. This is in contrast to the currently used greedy selection criteria based on performance metrics, which ignore the potential of the model to improve self-improving abilities.

We estimate CMP with the **weighted** average of agents' empirical performance in the clade. (See below for how our evaluation policy promotes more accurate estimation of CMP.) Formally, let us assume a fixed archive of agents $\mathcal{T}_t$, $n_{\text{success}}(a)$ be the number of passed tests of $a$, and $n_{\text{failure}}(a)$ be the number of failed tests of $a$. Then

$$n_{\text{success}}^C(a) = \sum_{a' \in C(a)} n_{\text{success}}(a') \quad \text{and} \quad n_{\text{failure}}^C(a) = \sum_{a' \in C(a)} n_{\text{failure}}(a').$$

Where $C(a)$ is the clade of $a$ in $\mathcal{T}_t$. We define our Clade-Metaproductivity estimator as

$$\widehat{\text{CMP}}(a) = \frac{n_{\text{success}}^C(a)}{n_{\text{success}}^C(a) + n_{\text{failure}}^C(a)},$$

Evaluating productivity at the level of entire clades rather than individual agents offers several key advantages. It aligns better with the goal of self-improvement, as a modest ancestor can still be highly valuable if its descendants consistently advance, while stagnant lineages are deprioritized. At the same time, aggregating evidence across a clade yields more statistically robust estimates than single-node outcomes by using information from more samples. This is particularly important when evaluations are costly and benchmarks are only partially observed.

$\widehat{\text{CMP}}(a)$ can be viewed as a weighted sum over empirical means of agents in $C(a)$, with the weight for an agent being the number of task evaluations it has. Furthermore, we design our evaluation

selection in such a way that it selects **highly performing agents**, which creates a selection of a soft maximum in the clade.

After calculating the CMP estimates, the HGM probabilistically approximates the selection of the highest scoring agent with Thompson Sampling - a standard method in the bandit literature for smoothly maximizing the decision criterion (Agrawal & Goyal, 2012; Chapelle & Li, 2011; Lattimore et al., 2020). We will refer to $a \sim TS(\{n_s, n_f | n \in \mathcal{T}_t\})$ as the agent sampled from the Thompson-Sampling process with parameters $n_s$ (number of successes) and $n_f$ (number of failures). Given the fact that the search problem has a known budget, our algorithm introduces an exploration-exploitation scheduler $\tau$ which is monotonically increasing with respect to the current time $t$, encouraging exploration in the early stage and polarization of the sampling distribution when approaching the end. Formally, we select the agent to expand $a^*$ as

$$a^* \sim TS(\{\tau(1 + n_{\text{success}}^C(a)), \tau(1 + n_{\text{failure}}^C(a)) | a \in \mathcal{T}_t\}).$$

**Evaluation Policy**  As stated in the expansion policy, we design our evaluation policy to prioritize **agents** with a higher evaluation score to induce the selection of the maximum over the clade. Formally, the agent to evaluate $a^*$ is sampled from the Thompson Sampling process with

$$a^* \sim TS(\tau(1 + n_{\text{success}}(a)), \tau(1 + n_{\text{failure}}(a)).$$

**Selection Policy**  Finally, our agent has to choose between expansion and evaluation. At each iteration, the algorithm first selects whether to evaluate or expand. Previous methods have evaluated newly created agents directly after their creation. Our novel estimation of agent self-improving quality has an additional benefit of collecting more samples faster (because it has samples from the entire clade). This enables a more fine-grained control over when to evaluate and when to create a new agent for better efficacy. Therefore, we decouple evaluation from expansion and treat them as separate steps.

To decide how and when to evaluate or expand agents, we draw inspiration from the infinite-armed bandit literature. Infinite-armed bandit problems capture the tension between repeatedly sampling known options to reduce uncertainty about promising arms and exploring new options that have the potential to perform better. This perspective provides a natural lens for our setting, where evaluations correspond to sampling existing arms and expansions correspond to introducing new ones. In this work, we follow the strategy of UCB-Air (Wang et al., 2008), which adds arms when the number of evaluations $N^\alpha \geq m$ for some $\alpha \in [0, 1]$, where $m$ is the number of existing arms. In our case, arms correspond to the agents; hence, we decide to expand at time $t$ if $N_t^\alpha \geq |\mathcal{T}_t|$.

**Final Agent Selection Strategy**  HGM iteratively executes the structured policy defined by our selection policy, expansion policy, and evaluation policy. When the computational budget exceeds, it returns the agent with the highest $\epsilon$ percentile of the utility posterior in the final tree for some hyperparameter $\epsilon$, namely the **best-belief agent**. Formally, a best-belief agent is defined as

$$\text{argmax}_{a \in \mathcal{T}_B} I_\epsilon(1 + n_{\text{success}(a)}, 1 + n_{\text{failure}}(a)),$$

where $I$ is the regularized incomplete beta function. See Algorithm1 in Appendix B for the detailed procedure of HGM.

**Asynchronous Implementation**  As an additional benefit of decoupling the policy, we introduce asynchronous execution of evaluation and expansion. Since the execution of coding agents generally requires querying large language models multiple times, the computation time can be lengthy. To boost our algorithm, we propose the asynchronous HGM algorithm (HGM Async), which utilizes all possible computational power until the computational budget is exceeded. HGM Async simultaneously executes one iteration process on each available CPU. Once one iteration finishes, a new iteration immediately starts. It uses the most recent data with one exception and updates the data once it finishes. The exception is that one needs to take all running expansions and explorations into consideration when executing the selection strategy. See experimental results 2 for run time comparison with DGM and SICA.

## 4 EXPERIMENTAL RESULTS

We evaluate HGM on challenging software engineering tasks to assess three core aspects: **1)** the fidelity of HGM's CMP estimation (Sec. 4.1), **2)** its capability for self-improvement with HGM com-

Table 1: **Clade-Metaproductivity: Empirical vs. Estimation Correlation**. We report the Pearson correlations between the empirical $\mathrm{CMP}s$ and the estimates from DGM, SICA, and HGM on SWE-Verified-60 and Polyglot. For the weighted correlations, each prediction is weighted by its accessed number of evaluations.

| Estimates | SWE-Verified-60 | | Polyglot | |
|---|---|---|---|---|
| | **Weighted** | **Un-weighted** | **Weighted** | **Un-weighted** |
| SICA | 0.444 | 0.444 | 0.274 | 0.274 |
| DGM | 0.285 | 0.406 | 0.383 | 0.357 |
| **HGM (Ours)** | **0.778** | **0.512** | **0.626** | **0.873** |

pared with DGM and SICA (Sec. 4.2), and **3)** the effectiveness in automatic agent design through evolutionary processes, benchmarked against a leading human design up to date[1] (Sec. 4.3). We conducted our experiments on the SWE-bench Verified (SWE-Verified) and SWE-bench Lite (SWE-Lite) variants, and the Polyglot problems, both consisting of coding challenges and are widely used for coding agent evaluation (Xia et al., 2025; Zhang et al., 2024; 2025b). We follow DGM's evaluation setting of Polyglot problems, where agents have no access to private test cases as well as test results. For budget considerations, in addition to the full datasets, we use 60-task subsets (SWE-Verified-60), derived from the first two stages of DGM's progressive evaluation. In all experiments, we employ HGM with an exploration-exploitation scheduler $\frac{B}{b}$, where $b$ is the remaining budget, $\epsilon = 1$, and $\alpha = 0.6$. All experiments involving HGM use the HGM-Async algorithm. We apply an identical initial agent when compared to DGM and SICA, which is adopted from the official implementation of DGM. See Appendix C.1 for a detailed description of the initial agents used in different experiments.

## 4.1 METAPRODUCTIVITY-PERFORMANCE MISMATCH

The experiments in this section are designed to serve two purposes: (i) to provide evidence of the Metaproductivity-Performance Mismatch (MPM) issue; and (ii) to assess whether the $\widehat{\mathrm{CMP}}$ of HGM is a more reliable CMP estimator than the utility measures adopted by DGM and SICA. To reveal the mismatch inherent in such reliance, we compute the correlation between their predictions and empirical CMP. To obtain empirical $\mathrm{CMP}s$, we analyze the expanded search tree after each method has completed its run. For every node in the tree, we define its empirical CMP as the maximum empirical mean of the task performance achieved within its clade with the root of this clade excluded. This construction ensures that empirical CMP captures the self-improvement ability of a node. We exclude the root of a clade to avoid circular use of the target in the estimators. For HGM, the $\widehat{\mathrm{CMP}}$ is defined as a function over the clade of a node; a critical adjustment is required to avoid target leakage. Specifically, we exclude the evaluations that are most directly related to the target: the root of the clade (an ancestor of the target) and the subtree rooted at a direct child of the evaluated node that contains the empirical maximum, thereby ensuring a fair comparison (see Appendix D for detailed computation). We report both the correlation coefficient weighted by the number of evaluations used in prediction, as well as the unweighted correlation. We conducted our experiments on the SWE-Veified-60 and Polyglot datasets.

**Results &. Discussion** Table 1 summarizes the correlations between the three estimators and the empirical $\mathrm{CMP}s$ (the targets). We first observe that the SICA and DGM estimators achieve positive Pearson correlation coefficients: 0.444 and 0.285 on SWE-Verified-60, and 0.274 and 0.383 on Polyglot, respectively, suggesting weak alignments, i.e., MPM. In comparison, HGM's estimator, $\widehat{\mathrm{CMP}}$, achieves substantially stronger weighted correlations of 0.778 and 0.626 on SWE-Verified-60 and Polyglot, respectively, as well as 0.512 and 0.873 unweighted correlation, surpassing SICA and DGM by wide margins. These results provide strong indications that HGM, equipped with $\widehat{\mathrm{CMP}}$, offers a more reliable estimate of metaproductivity and effectively mitigates the MPM issues inherent to SICA and DGM.

---

[1] The leading SWE-agents on `https://www.swebench.com` (Lite) as of 22 September 2025.

Table 2: **Self-Improving Capability Comparison**. We report the task performance (in accuracy) of each method's best-belief agent and the allocated CPU-hours time required for 800 evaluations. Super-scripted accuracies with "+" indicate performance gains over their respective initial agents.

| Best-belief Agent of | SWE-Verified-60 | | Polyglot | |
|---|---|---|---|---|
| | Acc. (%)↑ | Time (hours)↓ | Acc. (%)↑ | Time (hours)↓ |
| SICA | $50.0^{+10}$ | infinite loop | $25.4^{+5.1}$ | 572 |
| DGM | $53.3^{+13.3}$ | 1231 | $27.1^{+6.8}$ | 2385 |
| **HGM (Ours)** | $\mathbf{56.7^{+16.7}}$ | **517** | $\mathbf{30.5^{+10.2}}$ | **347** |

## 4.2 EVALUATING HGM SELF-IMPROVING CAPABILITY

To validate our hypothesis that our CMP estimator better predicts future self-improvement and hence leads to more effective self-modifications, we evaluate HGM against two state-of-the-art self-improving coding agent methods: DGM and SICA. We conduct controlled experiments under the same setup as DGM, with a budget-friendly modification applied consistently to both HGM and all baselines to ensure fairness, i.e., we adopt more cost-efficient backbone LLMs (GPT-5 for expansion and GPT-5 mini for evaluation for SWE-Verified; Qwen3-Coder-480B-A35B-Instruct for expansion and Qwen3-Coder-30B-A3B-Instruct for evaluation for Polyglot). For all methods, we start with the same initial ancestor, which achieves 40% and 20.3% on the SWE-Verified-60 and Polyglot, respectively. We compare the task performance of their best-belief final agents after a maximum allowance of 800 benchmark task evaluations, selected in DGM and SICA using empirical means. In addition, our asynchronous parallelization of expansion and evaluation should enable the self-improvement to consume fewer allocated CPU-hours than DGM and SICA (see Sec. 3.2). To verify this, we also report the allocated CPU-hours required for 800 evaluations.

**Results &. Discussion** We summarize the comparison results in Table 2. Across both SWE-Verified-60 and Polyglot, all three methods successfully perform agent discovery by optimizing the initial agent through self-improvement. However, HGM's best-belief agent demonstrates not only the highest task performance—56.7% on SWE-Verified-60 and 30.5% on Polyglot—but also the best efficiency, requiring the fewest allocated CPU-hours for 800 evaluations: 6.86× faster than DGM and 1.65× faster than SICA on Polyglot, and 2.38× faster than DGM on SWE-Verified-60. Notably, on SWE-Verified-60, SICA repeatedly encounters "query length out-of-LLM-context-window" during self-improvement processes after 360 evaluations. Despite this, the Polyglot results validate our hypothesis on HGM's runtime advantage over the baselines. In conclusion, HGM, equipped with a better utility estimator and asynchronous expansion-evaluation iterations, establishes itself as a more effective self-improving mechanism compared to DGM and SICA.

## 4.3 HGM VS. HUMANS: ON CODING AGENTS DESIGN

To gain a better understanding of its potential, we extend our evaluation of HGM by benchmarking it against the best human performance in coding agent design on SWE-Lite. We consider two settings: 1) optimization on full SWE-Verified and 2) generalization to SWE-Lite.

### 4.3.1 OPTIMIZATION ON FULL SWE-BENCH VERIFIED

In this experiment, rather than using the SWE-Verified-60, we scale HGM evaluation to the full SWE-Bench Verified benchmark (500 coding challenges) with an increased number of HGM iterations (8000 evaluations). Under this setup, the initial GPT-5 mini agent achieves 53.2% accuracy. Notably, this stronger starting point underscores the difficulty of further improvement: as task complexity grows and the search space expands, naive strategies tend to plateau.

**Results & Discussion** After 8000 evaluations, HGM discovers an agent that solves 61.4% of tasks, surpassing the best human-designed agent built on GPT-5 mini on the SWE-Verified leaderboard. This establishes our discovered agent as the *top-scoring* GPT-5 mini-based system, and positions it among the *top-10* agents over all checked submissions, even compared to systems built on stronger backbone models that can cost 5× more (e.g., Claude-3.7). While higher scores on the leaderboard do not necessarily indicate superior general coding ability—since both human- and machine-

Table 3: **Generalization on SWE-Lite and GPT-5.** We report the accuracy of HGM's best-belief SWE-Verified agent on SWE-Lite with GPT-5 mini and GPT-5 under two settings: filtered (completely unseen) and standard (leaderboard setting).

| Coding Agents | SWE-Lite Filtered (%) | SWE-Lite Standard (%) |
|---|---|---|
| SWE-agent + GPT-5 mini | 39.6 | 47.6 |
| **HGM's Best-belief Agent + GPT-5 mini** | **40.1** | **49.0** |
| SWE-agent + Claude 4 Sonnet (Best on the LB) | 48.3 | 56.7 |
| **HGM's Best-belief + GPT-5** | **48.8** | **57.3** |

designed agents may overfit to the benchmark—these results demonstrate a promising potential of HGM for competing with established human-designed baselines under identical model constraints.

### 4.3.2 GENERALIZATION TO DIFFERENT LLMS AND DATASETS

To ensure HGM improves general coding ability rather than overfitting to SWE-Verified, we evaluate the best-belief agent on SWE-Lite (300 tasks, 93 of which overlap with SWE-Verified). To isolate agent design from backbone effects, we compare the best-belief agent and SWE-agent, the leading system (with checked submissions), both using GPT-5 mini. We also examined how the discovered agent scales when paired with larger and better-performing LLMs by replacing the GPT-5 mini backbone with the GPT-5 model. Performance is reported in two settings: **Filtered** (completely unseen tasks) and **Standard** (leaderboard setting).

**Results & Discussion.** As shown in Table 3, HGM's best-belief agent, discovered on SWE-Verified, generalizes strongly to SWE-Lite, achieving $40.1\%$ in the **Filtered** setting and $49.0\%$ in the **Standard** setting. Compared to its initial ancestor ($34.8\%$ and $44.0\%$, respectively), these gains demonstrate that HGM's self-evolution improves general coding capability rather than overfitting to the optimization benchmark. Under identical GPT-5 mini backbones, the HGM-optimized agent outperforms the SWE-agent (which attains $39.6\%$ filtered and $47.6\%$ standard), indicating that the improvement stems from agent design rather than model choice. Furthermore, when replacing GPT-5 mini with GPT-5, the evolved agent maintains and further improves performance, surpassing all officially verified submissions on SWE-Bench Lite, confirming that HGM's self-evolved design principles transfer robustly across backbone scales and are not tied to a specific LLM.

**SWE-Bench-Live Result.** Due to recent contamination concerns regarding SWE-Bench (Zhang et al., 2025b), we also evaluate HGM's best-belief agent on a more recent benchmark, SWE-Bench-Live. Our agent with GPT-5.1 Codex mini achieves state-of-the-art performance across methods from the leaderboard, with an accuracy of 27.0% on SWE-Bench-Live Lite, outperforming the previous leading score of 24.7% on the leaderboard at the time of submission. It further verifies the strength of HGM and the transferability of its optimized agents to more up-to-date tasks and LLMs.

## 5 RELATED WORKS

The general concepts of machine self-improvement were first systematically articulated by Good (1966), who described the possibility of "Intelligence Explosion" once machines acquire the capacity to design more capable successors. Early work on explicit self-improvements dates back to Schmidhuber (1987), which introduced self-referential learning mechanisms in which a system generates and evaluates modified descendant versions of itself. Follow-up work on self-improvement progressed through interaction and agentic reinforcement learning. The Success-Story Algorithm(SSA) (Schmidhuber & Zhao, 1996; Schmidhuber et al., 1997) progressively forces self-modifying policies to discover more effective self-modification strategies. Its core mechanism is based on hindsight: at each checkpoint, a sequence of self-modifications that did not yield higher long-term reward rates is systematically undone. In this way, SSA enforces continual improvement by ensuring that only those self-modifications associated with demonstrably greater reward intake per unit time are preserved. Fitness-Monotonic Execution (Kirsch & Schmidhuber, 2022a;b) reduces the outer-loop design by favoring the execution of models with higher ancestral performance. Meta-discovered update rules optimized optimizers (Metz et al., 2021) and black-box search (Lange et al., 2023). On the other hand, the Gödel Machine, a fully self-referential algorithm that rewrites its

own code whenever it can *prove* an expected-utility improvement, provides a provably and globally optimal mechanism for self-improvement (Schmidhuber, 2003).

The rise of contemporary LLMs has created an opportunity to automate substantial aspects of software engineering. One concrete step in this direction is the development of coding agents, which extend LLMs with the ability to operate in conventional computing environments. ChatDev (Qian et al., 2023) first illustrated this idea in the context of automated bug fixing, and similar frameworks were later explored in SWE (Yang et al., 2024), OpenHands (Wang et al., 2024), MetaGPT (Hong et al., 2024), and AgentLess (Xia et al., 2025).

The Self-Taught Optimizer (Zelikman et al., 2024) and Gödel Agent (Yin et al., 2024) first experimented with agents that modify their own scaffolding. Subsequently, DGM (Zhang et al., 2025a) and SICA (Robeyns et al., 2025) extend this direction by implementing self-modifying machines as full software engineering projects, where agents modify their own repositories while validating changes through execution-grounded software engineering tasks. Both DGM and SICA, explicitly or implicitly, assume that higher software benchmark scores correspond to greater self-improvement capacity. In contrast, HGM introduces a qualitative measure of self-improvement consistent with the theoretical Gödel Machine and directs self-modifications using estimates of this measure.

The identified tree-search problem spans fixed-budget best-arm identification (BAI), Monte Carlo Tree Search, and infinite-armed bandits, introducing a distinct decision: explicit expansion actions that create new candidate leaves alongside ordinary evaluations. Fixed-budget BAI and Bayesian value-of-information methods assume a finite and known set of arms and offer guarantees for static candidates, thus not modeling the discovery of unknown arms (Audibert & Bubeck, 2010; Karnin et al., 2013; Frazier et al., 2008). Monte-Carlo Tree Search and its UCT variants (Coulom, 2006; Kocsis & Szepesvári, 2006) alternate selection, expansion, and simulation, while their backup and selection rules typically target cumulative reward rather than fixed-budget final-choice objectives under noisy, low-signal feedback, with limited guarantees for pure exploration of leaf quality (Kaufmann & Koolen, 2017). Infinite-armed bandit formulations capture the explore-discover tradeoff but typically model discoveries as i.i.d. draws from a reservoir, missing tree structure, and hierarchical dependencies (Wang et al., 2008; Bubeck et al., 2011; Carpentier & Valko, 2015).

## 6 CONCLUSION

In this work, we identify a key limitation in the search heuristics of current self-improving coding agents: benchmark scores alone do not reliably reflect an agent's long-term self-improvement potential, as high-scoring agents may yield stagnating lineages while weaker ones can seed productive improvements. We term this the Metaproductivity-Performance Mismatch. To address it, we introduce Clade-Metaproductivity (CMP), a lineage-based metric inspired by Huxley's notion of clades. We show that, under Assumption 1, when applied to our self-improving coding agent search problem, a CMP oracle is sufficient to implement the Gödel Machine (Theorem 1).

Building on this principle, we propose the Huxley-Gödel Machine (HGM), which approximates CMP and guides expansion via Thompson sampling with adaptive scheduling. Empirically, HGM consistently outperforms previous self-improving methods while reducing allocated CPU hours. It achieves human-level coding agent design performance on SWE-bench Lite with GPT-5, despite being optimized on SWE-bench Verified with GPT-5 mini, demonstrating generalization across datasets and model shifts. The HGM discovered agent also achieves state-of-the-art performance on SWE-bench-Live Lite when paired with GPT-5.1 Codex mini. Together, these results indicate that our clade-based measure of improvement potential, rather than immediate benchmark performance, enables more effective self-improvement, suggesting a paradigm in which agent improvement is driven by the long-term metaproductivity of entire lineages rather than short-term gains.

While HGM currently focuses on symbolic self-improvement, editing scaffolds, prompts, and high-level control logic while treating architectures and weights as fixed hardware, an interesting next step is extending this framework to operate directly over weight space. Weight-level self-modification (Schmidhuber, 1993; Metz et al., 2021; Irie et al., 2022; Kirsch & Schmidhuber, 2022b; Lu et al., 2024; Ishibashi et al., 2024) would open an even larger landscape of potential improvements. Although our present instantiation remains at the symbolic layer, the underlying mechanism is intrinsically compatible with weight-level modifications. We view this integration of symbolic and parametric self-improvement as a promising and ambitious direction for future work.

ACKNOWLEDGMENT

We thank Yuhui Wang for the discussions during the early stages of this project. We gratefully acknowledge Jenny Zhang and Shengran Hu, the authors of Darwin Gödel Machine, for sharing their insights about DGM and their implementation experience. We also thank Yilan Zhang, Rui Zhang, and Lisiyu Xie for their help in designing the visualizations. The research reported in this publication was supported by funding from King Abdullah University of Science and Technology (KAUST) - Center of Excellence for Generative AI, under award number 5940.

ETHICS STATEMENT

An agent that can rewrite itself arbitrarily carries inherent risk, i.e., the agent may eventually change in unexpected ways. We do the necessary due diligence to ensure that the HGM's behavior is performed in a safe, fully sandboxed environment. This ensures that any self-modifications remain contained and cannot influence systems beyond the experimental setup. More importantly, studying self-improving agents in such controlled settings allows us to probe how these systems evolve, what kinds of changes they gravitate toward, and how their capabilities unfold over time. By observing the dynamics of self-modification at this early stage, we gain insight into the mechanisms that might govern more advanced systems in the future. Moreover, our CMP and its estimations provide a framework in which the dynamics of self-modifying agents can be examined empirically and theoretically. We believe this framework paves a conceptual foundation for understanding, guiding, and ultimately designing safer self-improving AI.

REPRODUCIBILITY STATEMENT

The codebase to produce our experimental results is based on the officially released GitHub Repository of Darwin Gödel Machine Zhang et al. (2025a). Adjustments have been made as detailed in Appendix C.1. We provide our source code to reproduce the results as reported in Section 4 in the supplementary material.

LLM USAGE

Large language models (LLMs) are used to help discover relevant works in the literature. In particular, they have been used for suggesting papers in upper-bound based tree search algorithms. LLMs are also used to polish the writing and fix grammatical errors.

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

## A  Gödel Machine with CMP Oracle

The proof of Gödel Machine being simulated with a $\mathrm{CMP}_\pi$ oracle is a simple observation that in the Gödel Machine setup, $\mathrm{CMP}_\pi$ is an actual state-action value function $Q_\pi$ (Sutton et al., 1998) defined by the Gödel Machine, as the agents above the clades are not reachable according to the design of Gödel Machines. In order to state this, we formalize each term precisely.

We start by formulating the assumptions and definitions precisely, restating the theorem and then we write down the proof.

*Assumptions.* The original Gödel Machine is defined in a time-aware setting, where the prover must establish not only that a proposed self-modification increases expected objectives, but also that this improvement still holds after accounting for the time required to search for proofs and compute the modification. This is necessary because, in the general case, the environment may change during these computations, and the objective is measured with respect to elapsed time.

In our setup, by contrast, the environment and the evaluation metric remain fixed throughout the agent's execution. The benchmark does not evolve over time, and the utility of any given agent is determined solely by its final performance on this static task. Importantly, we assume that the utility is measured by evaluation on tasks. It also follows the assumption of repeatable trials, meaning that the evaluation of a given agent on a task is independent of evaluation time or prior events. In other words, we are able to reset the testing environment for each test. Furthermore, we assume that the Gödel Machine prover has full knowledge of the utility function as part of its axioms of the environment. Hence, we exclude the evaluation actions from the action space. Finally, we assume that the Gödel Machine prover does not consume budget, and that the self-modifications consume an equal amount of budget—exactly one budget unit.

We show that the CMP oracle is sufficient to imitate Gödel Machines in our specific setting that satisfies the following:

- The policy objective function is defined as a function of only the final agent, with no other rewards received before termination;

- The agent's utility is measured by its performance on evaluation tasks, under the assumption of repeatable trials: for any agent-task pair, the expected outcome is independent of evaluation time or prior events;

- The Gödel Machine operates in a complete axiomatic system with full knowledge of the utility function;

- The computation budget is finite and known;

- The proofs of Gödel Machines do not consume the budget;

- And each self-modification costs exactly one unit of the budget.

Formally, we define the Gödel Machine as an optimal policy on the POMDP (let's call it Gödel POMDP) defined as:

**State space.**  Gödel POMDP operates on an extended state space, which, in addition to the tree of agents $\mathcal{T}$, consists of two special agents: $a_{\mathrm{parent}}$ and $a_{\mathrm{child}}$, from the tree $\mathcal{T}$. For a full formal description, let's also include a remaining budget descriptor that determines the number of transitions left before the budget is exceeded.

**Observation space.** Observation is limited to the agents $(a_{\mathrm{parent}}, a_{\mathrm{child}})$ and the remaining budget $b$.

**Action space.**  For most states, the action space consists of two actions—accept and reject. As in the original Gödel Machine, the action selects the new parent in the next iteration. Intuitively, accept selects $a_{\mathrm{child}}$ as the new parent, and reject requires the parent to remain unchanged. As the policy operates on a POMDP, it is a function of the history of the observations, which is equivalent to the state—tree $\mathcal{T}$ with special states $a_{\mathrm{parent}}$ and $a_{\mathrm{child}}$. For the sake of simplicity in notation, we will refer to the newly selected parent agent as if it were the action. Thus, for policy $\pi$, $a_{\mathrm{parent}} = \pi((\mathcal{T}, a_{\mathrm{parent}}, a_{\mathrm{child}}, b))$ corresponds to the action reject.

**Scoring function.** Each policy also consists of a $Score_\pi$ function that scores all possible elements in the observation. Formally, $Score_\pi : Obs \rightarrow \mathcal{T} \rightarrow \mathbb{R}$ is a function from observation into a function of agents that is returned when the budget $b$ is equal to 0, from which the agent with the highest score is selected as the final agent. It is a function of the observation and not the entire history, as the algorithm can only choose the final agent from the final observation.

The policy selects the highest-scoring agent, as determined by the $Score_\pi(o)$ function. Furthermore, the original Gödel Machine at the end allows only the parent or current child to be kept as the final agent. Hence, we restrict the score functions such that they always produce indicator functions of one of the observed agents, i.e., one observed agent has a score of 1, and all other agents receive a score of zero. The scoring function corresponds to the selection made by the algorithm in our original setup. After the selection dictated by the scoring function, the agent is evaluated by the Utility function $U$.

**Transitions.** Transitions generate new self-modifications from the newly selected next parent, i.e., the action output. At each transition, a self-modification of $a_{\text{parent}}^{\text{new}}$ is created and assigned as a new node $a_{\text{child}}$. $a_{\text{parent}}^{\text{new}}$ becomes the $a_{\text{parent}}$ for the next time step. Formally, if $a(a)$ is the self-modification of $a$, and $Tr$ is the transition function, then $Tr((\mathcal{T}, a_{\text{parent}}, a_{\text{child}}, b), a) = (\mathcal{T} \cup \{a(a)\}, a, a(a), b-1)$. The final reward in Gödel POMDP is $U(\arg\max_{a \in \mathcal{T}_B} Score_\pi(a_{\text{parent}}, a_{\text{child}})(a))$.

Theorem 1 now can be formally restated:

---

**Theorem 1.** *Access to a* CMP *oracle is sufficient to implement a Gödel Machine.*

---

*Proof.* For the proof we adapt the original coding setting to match the notation of Gödel POMDP. Let's call it the Extended MDP. Similarly to Gödel POMDP, its state is a tree $\mathcal{T}$, involving two special agents $a_{\text{parent}}, a_{\text{child}}$ and the budget $b$. The action space is the same as in our problem and consists of a single agent $a \in \mathcal{T}$ when there is still some budget left. In other cases, the policy $\pi$ uses its $Score_\pi$ function to score created agents in the trees for the selection of the final agent. The transition function adds the self-modification to the tree and sets $a$ as the new $a_{\text{parent}}$ and $a(a)$ as the new $a_{\text{child}}$.

Every policy from Gödel POMDP can be mapped to a policy in the extended MDP. Specifically, let $\pi_G$ be a policy in the Gödel POMDP, then we define the corresponding policy in the extended MDP $\pi_E$ as $\pi_E((\mathcal{T}, a_{\text{parent}}, a_{\text{child}}, b)) = \pi_G((\mathcal{T}, a_{\text{parent}}, a_{\text{child}}, b))$. At each node, the policy chooses between the two special nodes, ignoring the rest of the tree.

Now we adapt CMP to Gödel POMDP. CMP adapted to the Gödel case is still a function of a state (and agent to expand) in the extended state space (not only observation) and the agent.

$$
\begin{aligned}
\text{CMP}_\pi((\mathcal{T}, a_p, a_c, b), a) &= \mathbb{E}_{(\mathcal{T}_B, a_{Bp}, a_{Bc}, 0) \sim p_\pi(\cdot | (\mathcal{T}, a_p, a_c, b), a)} \left[ U(\arg\max_{a' \in C(\mathcal{T}_B, a)} Score_\pi(a_{Bp}, a_{Bc})(a')) \right] \\
&\overset{(2)}{=} \mathbb{E}_{(\mathcal{T}_B, a_{Bp}, a_{Bc}, 0) \sim p_\pi(\cdot | (\mathcal{T}, a_p, a_c, b), a)} \left[ U(\arg\max_{a' \in \{a_{Bp}, a_{Bc}\}} Score_\pi(a_{Bp}, a_{Bc})(a')) \right] \\
&\overset{(3)}{=} Q_\pi((\mathcal{T}, a_p, a_c, b), a).
\end{aligned}
$$

Where $\mathcal{T}_B$ is the final unrolled tree, $a_{B_p}$ is the final parent and $a_{B_c}$ is the final child in the sampled trajectory. The second equality (2) comes from the fact that the $Score_\pi$ is equal to 1 for exactly one element from the final observation -either $a_{B_p}$ or $a_{B_c}$ (paragraph A). The third equality (3) comes from directly unrolling the $Q_\pi$-value function.

The Gödel Machine is defined by a prover that produces a proof whether accepting or selecting a given node (or rejecting). Here, the algorithm by having access to CMP has access to the true $Q$ value function in the Gödel POMDP. This serves as a proof of better objective of either parent or the child. Additionally, we state that we break the tie by selecting the parent node as in the original Gödel Machine. Hence, the algorithm that follows it is a Gödel Machine.

Independently, as this proof directly shows that Gödel Machine selects an action that maximizes its own $Q_\pi$-value function, it is optimal due to the Bellman Optimality Equation. With the procedure shown above, we can adapt it to the extended MDP.

$\square$

## B  ALGORITHM

Algorithm 1 presents the procedure of HGM.

In the standard HGM, when an expansion ends with a new $a$ added to the tree, the next following evaluation actions have the choice of choosing $a$. At the beginning of the search, the first actions have the most dense ratio of expansion actions, and the following evaluation actions would diverge more from the standard HGM since most of their choices are not generated yet, concentrating on the early finished expansions. Therefore, to avoid this bias, we initialize by expanding the initial agent 5 times with each of the processes in parallel.

The asynchronization also introduces another bias that favors agents with fewer evaluated results to be more often selected for evaluation than in the standard HGM. This is due to the fact that the easy tasks usually stop earlier than the difficult ones. This yields agents with fewer evaluations having a higher empirical mean. During our experiments, we observed that many agents were successful on the first ten evaluated tasks; however, this accuracy drops quickly as the number of evaluations grows. After having more than 50 evaluations, this bias is barely observed.

---

**Algorithm 1** Huxley–Gödel Machine (HGM)

---

1: **Input:** the initial agent $a_0$, widening parameter $\alpha$, and the percentile $\epsilon$ for final selection
2: Initialize a tree $\mathcal{T}$ with root $a_0$
3: Initialize counters $n_{\text{success}}^C(a), n_{\text{failure}}^C(a), n_{\text{success}}(a), n_{\text{failure}}(a)$ for all $a \in \mathcal{T}$
4: **while** Computational Budget not Exceeded **do**
5:    **if** $|\mathcal{T}| \leq n^\alpha$ and expandable parents exist **then**
6:       **Expand:**
7:       **for** each node $a \in \mathcal{T}$ **do**
8:          Sample $S_C(a) \sim \text{Beta}(\tau(1 + n_{\text{success}}^C(a)),\ \tau(1 + n_{\text{failure}}^C(a)))$
9:       **end for**
10:      Select node $a^\star = \arg\max_a S_C(a)$
11:      Create child $c$ by self-modification of $a^\star$
12:      Add $c$ to $\mathcal{T}$
13:    **else**
14:       **Evaluate:**
15:       **for** each agent $a \in \mathcal{T}$ with remaining tasks **do**
16:          Sample $S(a) \sim \text{Beta}(\tau(1 + n_{\text{success}}(a)),\ \tau(1 + n_{\text{failure}}(a)))$
17:       **end for**
18:      Select agent $a^\star = \arg\max_a S(a)$
19:      Allocate a benchmark task to $a^\star$
20:      update $n_{\text{success}}, n_{\text{failure}}$ for $a^\star$
21:      update $n_{\text{success}}^C, n_{\text{failure}}^C$ for $a^\star$ and ancestors
22:    **end if**
23: **end while**
24: **Return** $\text{argmax}_{a \in \mathcal{T}} I_\epsilon(1 + n_{\text{success}(a)}, 1 + n_{\text{failure}}(a))$

---

## C  EXPERIMENTAL DETAILS

### C.1  INITIAL AGENTS

Our initial agents applied in Section 4.2 are adopted from the official implementation of DGM with minor changes, including modifying API support, setting up a timeout option, and adding a length of LLM interaction restriction. The initial agent is essentially a single loop of LLM queries with two tool options, i.e., file editing and bash command execution. We set a time limit of one hour for each agent execution.

The initial agents used in SWE-bench experiments and Polyglot experiments differ in that the Polyglot initial agent includes test commands with different programming language support. There are two additional functions in the SWE-bench initial agent that serve to summarize existing tests and execute the tests with a report generated, respectively.

The initial agent employed in Section 4.3 is further adjusted by removing the file-editing tool, leaving only the bash tool, to minimize initial inductive bias. The time limit is extended to five hours for both self-modification and task evaluation, reducing the risk of prematurely eliminating stronger agents due to time constraints.

### C.2  OTHER DETAILS

For the Polyglot experiments presented in Section 4.2, the exact large language model used for self-modification is an int4 and int8 mixed quantized version of Qwen3-Coder-480B-A35B-Instruct generated by AutoRound (Intel, 2025). Overall, we spent approximately $5000 USD to produce the experimental results, including all three methods.

## D   EMPIRICAL CMP AND ITS ESTIMATION

In this section, we provide the exact formula to compute the empirical CMP and the variant of our CMP estimator being used in Section 4.1 for correlation analysis. The empirical CMP of an agent $a$ as a node in a tree is defined as

$$\max_{a' \in C(a) \setminus \{a\}} \frac{n_{\text{success}}(a')}{n_{\text{success}}(a') + n_{\text{failure}}(a')}.$$

The prediction of our CMP estimator is defined as

$$\frac{n_{\text{success}}^{C}(a) - n_{\text{success}}(a) - n_{\text{success}}^{C}(b^*)}{n_{\text{failure}}^{C}(a) - n_{\text{failure}}(a) - n_{\text{failure}}^{C}(b^*) + n_{\text{success}}^{C}(a) - n_{\text{success}}(a) - n_{\text{success}}^{C}(b^*)},$$

where $b^*$ is a child of $a$ such that

$$\left( \text{argmax}_{n \in C(a)} \frac{n_{\text{success}}(n)}{n_{\text{failure}}(n)} \right) \cap C(b^*) \neq \emptyset.$$

For both SICA and DGM, we consider the benchmark performance of an agent as their estimator of the agent's CMP.

# E  BASELINES

Table 2 summarizes the three subpolicies of SICA, DGM, and HGM, which define solutions to the iterative tree search problem defined in 2.

| Subpolicy | SICA | DGM | HGM (Ours) |
|---|---|---|---|
| **Mod vs. Eval** | Alternates between modification and evaluation. | Alternates between modification and evaluation. | Adaptive choice between modification and evaluation. |
| **Modification Strategy** | Greedily selects the agent with the best performance up to this point and modifies it with the entire history accessible to the agent. | Selects the node probabilistically based on the evaluation metric and the number of children of the agents. | Selects the node based on the statistics of the *clade* stemming from a given node. |
| **Evaluation Strategy** | Evaluates the most recently created agent on the entire evaluation dataset. | Progressively evaluates the last created agent on subsets of the dataset, expanding if results are promising. | Selects the agent based on the statistics and evaluates it on a single task. |

Table 4: Comparison of structured policies across self-improving methods. Each method is described by three subpolicies: Mod vs. Eval, Modification Strategy, and Evaluation Strategy.

## F  DISCOVERED AGENTS

We present interesting findings about HGM-discovered agents during our experiments. By manually inspecting the diff patch files that were generated by self-modification, we make interesting observations.

One engaging self-modification we found implements iterative refinement to make incremental improvements. In the self-improving context, it means that during one expansion (self-modification) step, the agent is accurately performing multiple self-modifications. More interestingly, this phenomenon can be viewed as the agent being self-motivated to perform self-modification, instead of being instructed to do so by humans. This self-motivated repetitive self-modification automatically creates arbitrary levels of meta-improvement in the sense that each self-modification ideally needs to consider all its subsequent self-modifications, which the subsequent self-modifications also need to consider future self-modifications. We show in List 1 the part of the patch file that includes the implementation of this self-modification.

Another interesting phenomenon we observe is the file structures that self-modifying agents create. Many agents tend to write the chunk they self-modify into a diff patch file, usually with a constant file name. When multiple consecutive attempts of these self-modifying actions apply, the structure of these git diff files becomes abstruse. Imagine that there is a diff file that includes certain edits. Once there are changes applied to this file, it then generates the diff patch file. In this new patch file, a nested format of editing record appears. At the beginning of some lines, there are multiple + and - signs that correspond to nested file edits. This is a natural format to present information that inspires long-sighted self-improvement since it shows how self-modifications indeed influence subsequent self-modifications. The authors find it mind-bending to understand such a file structure and parse the semantics manually. For readers interested in further investigation, we provide one instance of such a file structure in List 2.

Listing 1: The self-modifying agent that executes multiple self-modifications per instruction.

```
+def attempt_error_resolution(git_dir, test_output, test_error, language)
    ↪ :
+    """
+    Attempt to automatically diagnose and resolve errors.
+    Returns a tuple of (resolved, message) where resolved indicates if
    ↪ errors were fixed.
+    """
+    safe_log("Attempting automated error diagnosis and resolution...")
+
+    # Diagnose errors using our enhanced bash tool function
+    diagnosis = diagnose_errors(test_output, test_error, "")
+
+    if not diagnosis["has_errors"]:
+        return False, "No errors detected to resolve."
+
+    resolution_messages = []
+
+    # Try to apply automated fixes for each diagnosed error
+    for error in diagnosis["errors"]:
+        safe_log(f"Processing error: {error['type']} - {error['
    ↪ description']}")
+
+        # Simple resolution strategies based on error type
+        if error["type"] == "python_module_not_found":
+            # For Python module not found errors, we might install the
    ↪ module
+            match = re.search(r"No module named '([^']+)'", error["
    ↪ description"])
+            if match:
+                module = match.group(1)
+                resolution_messages.append(f"Would attempt to install
    ↪ Python module: {module}")
+                # In practice, we would run: pip install {module}
```

```
+                # But we'll skip actual installation to avoid side
    ↪ effects
+
+       elif error["type"] == "python_syntax_error" and "file" in error:
+           # For syntax errors, we could potentially apply fixes
+           file_path = os.path.join(git_dir, error["file"])
+           if os.path.exists(file_path):
+               resolution_messages.append(f"Would attempt to fix syntax
    ↪ error in {file_path} at line {error.get('line', 'unknown')}")
+               # In practice, we would use the editor tool's apply_fix
    ↪ command
+               # This is just a demonstration of what could be done
+
+       elif error["type"] == "test_failure":
+           # For test failures, we might suggest reviewing the
    ↪ implementation
+           resolution_messages.append("Would analyze test failures and
    ↪ suggest implementation improvements")
+
+   if resolution_messages:
+       return True, "Automated resolution attempted:\n" + "\n".join(
    ↪ resolution_messages)
+   else:
+       return False, "No automated resolutions available for detected
    ↪ errors."
+
class AgenticSystem:
    def __init__(
            self,
@@ -243,6 +293,16 @@ Your task is to make changes to the files in the {
    ↪ self.git_dir} directory to add

            safe_log(f"Attempt {attempt + 1} test results: {'PASSED' if
    ↪ test_success else 'FAILED'}")

+           # If tests failed, attempt automated error resolution
+           if not test_success:
+               resolved, resolution_message = attempt_error_resolution(
+                   self.git_dir, test_output, test_error, self.language
+               )
+               safe_log(f"Error resolution: {resolution_message}")
+
+               # Even if we couldn't automatically resolve, we still
    ↪ provide feedback
+               # In a more advanced implementation, we might actually
    ↪ apply fixes here
+
            # If this is the first attempt or tests passed and we didn't
    ↪  have a successful attempt yet, update best patch
            if attempt == 0 or (test_success and (best_patch is None or
    ↪ not best_test_results)):
                best_patch = current_patch
@@ -278,37 +338,31 @@ Please revise your code to fix these issues and try
    ↪  again.
        # Log final summary
        safe_log(f"\n{'='*20} FINAL SUMMARY {'='*20}")
        safe_log(f"Best solution found on attempt: {best_test_results['
    ↪ attempt'] if best_test_results else 'None'}")
-       safe_log(f"Tests passed: {best_test_results['test_success'] if
    ↪ best_test_results else 'Unknown'}")
+       safe_log(f"Final test result: {'PASSED' if best_test_results and
    ↪  best_test_results['test_success'] else 'FAILED'}")
+
+       if best_test_results:
```

```
+            safe_log(f"Final test output:\n{best_test_results['
    ↪ test_output']}")
+            if best_test_results['test_error']:
+                safe_log(f"Final test errors:\n{best_test_results['
    ↪ test_error']}")

-        # Save attempt history to a file
-        history_file = os.path.join(os.path.dirname(self.
    ↪ chat_history_file), 'attempt_history.md')
-        with open(history_file, 'w') as f:
-            f.write("# Attempt History\n\n")
-            for result in self.attempt_history:
-                f.write(f"## Attempt {result['attempt']}\n")
-                f.write(f"**Tests Passed**: {result['test_success']}\n")
-                f.write(f"**LLM Calls Used**: {result['llm_calls']}\n")
-                f.write(f"**Test Output**:\n```\n{result['test_output
    ↪ ']}\n```\n")
-                f.write(f"**Test Error**:\n```\n{result['test_error']}\n
    ↪ ```\n")
-                f.write(f"**Patch**:\n```\n{result['patch']}\n```\n\n")
+        return bool(best_test_results and best_test_results['
    ↪ test_success'])
```

Listing 2: An instance of the nested diff patch format.

```
diff --git a/attempt_history.md b/attempt_history.md
new file mode 100644
index 0000000..b132b1a
--- /dev/null
+++ b/attempt_history.md
@@ -0,0 +1,727 @@
+# Attempt History
+
+## Attempt 1
+**Tests Passed**: True
+**LLM Calls Used**: 18
+**Test Output**:
+```
+============================ test session starts
    ↪ =============================
+platform linux -- Python 3.10.18, pytest-8.4.2, pluggy-1.6.0 -- /usr/
    ↪ local/bin/python3.10
+cachedir: .pytest_cache
+rootdir: /dgm
+configfile: pytest.ini
+testpaths: tests
+plugins: asyncio-1.1.0, anyio-4.10.0
+asyncio: mode=strict, asyncio_default_fixture_loop_scope=None,
    ↪ asyncio_default_test_loop_scope=function
+collecting ... collected 29 items
+
+tests/test_bash_tool.py::TestBashTool::test_simple_command PASSED
    ↪ [  3%]
+tests/test_bash_tool.py::TestBashTool::test_multiple_commands PASSED
    ↪ [  6%]
+tests/test_bash_tool.py::TestBashTool::test_command_with_error PASSED
    ↪ [ 10%]
+tests/test_bash_tool.py::TestBashTool::test_environment_variables PASSED
    ↪ [ 13%]
+tests/test_bash_tool.py::TestBashTool::test_command_output_processing
    ↪ PASSED [ 17%]
+tests/test_bash_tool.py::TestBashTool::test_long_running_command PASSED
    ↪ [ 20%]
+tests/test_bash_tool.py::TestBashTool::test_invalid_commands[
    ↪ invalid_command_name] PASSED [ 24%]
```

```
+tests/test_bash_tool.py::TestBashTool::test_invalid_commands[cd /
    ↪ nonexistent/path] PASSED [ 27%]
+tests/test_bash_tool.py::TestBashTool::test_invalid_commands[/bin/
    ↪ nonexistent] PASSED [ 31%]
+tests/test_bash_tool.py::TestBashTool::test_command_with_special_chars
    ↪ PASSED [ 34%]
+tests/test_bash_tool.py::TestBashTool::test_multiple_line_output PASSED
    ↪ [ 37%]
+tests/test_bash_tool.py::TestBashTool::test_large_output_handling PASSED
    ↪ [ 41%]
+tests/test_edit_tool.py::TestEditorTool::test_view_file PASSED
    ↪ [ 44%]
+tests/test_edit_tool.py::TestEditorTool::test_create_file PASSED
    ↪ [ 48%]
+tests/test_edit_tool.py::TestEditorTool::test_create_existing_file
    ↪ PASSED [ 51%]
+tests/test_edit_tool.py::TestEditorTool::test_edit_file PASSED
    ↪ [ 55%]
+tests/test_edit_tool.py::TestEditorTool::test_edit_nonexistent_file
    ↪ PASSED [ 58%]
+tests/test_edit_tool.py::TestEditorTool::test_view_directory PASSED
    ↪ [ 62%]
+tests/test_edit_tool.py::TestEditorTool::test_invalid_path PASSED
    ↪ [ 65%]
+tests/test_edit_tool.py::TestEditorTool::test_invalid_commands[
    ↪ unknown_command] PASSED [ 68%]
+tests/test_edit_tool.py::TestEditorTool::test_invalid_commands[] PASSED
    ↪ [ 72%]
+tests/test_edit_tool.py::TestEditorTool::test_invalid_commands[None]
    ↪ PASSED [ 75%]
+tests/test_error_diagnosis.py::TestErrorDiagnosis::
    ↪ test_python_syntax_error_diagnosis PASSED [ 79%]
+tests/test_error_diagnosis.py::TestErrorDiagnosis::
    ↪ test_python_module_not_found_diagnosis PASSED [ 82%]
+tests/test_error_diagnosis.py::TestErrorDiagnosis::
    ↪ test_no_error_diagnosis PASSED [ 86%]
+tests/test_error_diagnosis.py::TestErrorDiagnosis::
    ↪ test_format_diagnosis_with_errors PASSED [ 89%]
+tests/test_error_diagnosis.py::TestErrorDiagnosis::
    ↪ test_format_diagnosis_without_errors PASSED [ 93%]
+tests/test_error_diagnosis.py::TestAutomatedFixes::
    ↪ test_apply_missing_import_fix PASSED [ 96%]
+tests/test_error_diagnosis.py::TestAutomatedFixes::
    ↪ test_apply_syntax_error_fix PASSED [100%]
+
+=================================== PASSES
    ↪ ====================================
+=========================== short test summary info
    ↪ ============================
+PASSED tests/test_bash_tool.py::TestBashTool::test_simple_command
+PASSED tests/test_bash_tool.py::TestBashTool::test_multiple_commands
+PASSED tests/test_bash_tool.py::TestBashTool::test_command_with_error
+PASSED tests/test_bash_tool.py::TestBashTool::test_environment_variables
+PASSED tests/test_bash_tool.py::TestBashTool::
    ↪ test_command_output_processing
+PASSED tests/test_bash_tool.py::TestBashTool::test_long_running_command
+PASSED tests/test_bash_tool.py::TestBashTool::test_invalid_commands[
    ↪ invalid_command_name]
+PASSED tests/test_bash_tool.py::TestBashTool::test_invalid_commands[cd /
    ↪ nonexistent/path]
+PASSED tests/test_bash_tool.py::TestBashTool::test_invalid_commands[/bin
    ↪ /nonexistent]
+PASSED tests/test_bash_tool.py::TestBashTool::
    ↪ test_command_with_special_chars
+PASSED tests/test_bash_tool.py::TestBashTool::test_multiple_line_output
```

```
+PASSED tests/test_bash_tool.py::TestBashTool::test_large_output_handling
+PASSED tests/test_edit_tool.py::TestEditorTool::test_view_file
+PASSED tests/test_edit_tool.py::TestEditorTool::test_create_file
+PASSED tests/test_edit_tool.py::TestEditorTool::
    ↪ test_create_existing_file
+PASSED tests/test_edit_tool.py::TestEditorTool::test_edit_file
+PASSED tests/test_edit_tool.py::TestEditorTool::
    ↪ test_edit_nonexistent_file
+PASSED tests/test_edit_tool.py::TestEditorTool::test_view_directory
+PASSED tests/test_edit_tool.py::TestEditorTool::test_invalid_path
+PASSED tests/test_edit_tool.py::TestEditorTool::test_invalid_commands[
    ↪ unknown_command]
+PASSED tests/test_edit_tool.py::TestEditorTool::test_invalid_commands[]
+PASSED tests/test_edit_tool.py::TestEditorTool::test_invalid_commands[
    ↪ None]
+PASSED tests/test_error_diagnosis.py::TestErrorDiagnosis::
    ↪ test_python_syntax_error_diagnosis
+PASSED tests/test_error_diagnosis.py::TestErrorDiagnosis::
    ↪ test_python_module_not_found_diagnosis
+PASSED tests/test_error_diagnosis.py::TestErrorDiagnosis::
    ↪ test_no_error_diagnosis
+PASSED tests/test_error_diagnosis.py::TestErrorDiagnosis::
    ↪ test_format_diagnosis_with_errors
+PASSED tests/test_error_diagnosis.py::TestErrorDiagnosis::
    ↪ test_format_diagnosis_without_errors
+PASSED tests/test_error_diagnosis.py::TestAutomatedFixes::
    ↪ test_apply_missing_import_fix
+PASSED tests/test_error_diagnosis.py::TestAutomatedFixes::
    ↪ test_apply_syntax_error_fix
+============================== 29 passed in 3.58s
    ↪ ==============================
+
+```
+**Test Error**:
+```
+
+```
+**Patch**:
+```
+diff --git a/coding_agent.py b/coding_agent.py
+index 78e8ad4..77e5097 100644
+--- a/coding_agent.py
++++ b/coding_agent.py
+@@ -5,9 +5,13 @@ from logging.handlers import RotatingFileHandler
+ import os
+ import threading
+ import time
++import json
++import re
+
+ from llm_withtools import CLAUDE_MODEL, OPENAI_MODEL, chat_with_agent
+ from utils.git_utils import diff_versus_commit, reset_to_commit,
    ↪ apply_patch
++from tools.bash import diagnose_errors
++from tools.edit import apply_automated_fix, read_file, write_file
+
+ # reset_to_commit(git_dname, commit)
+ # apply_patch(git_dname, patch_str)
+@@ -136,6 +140,52 @@ def run_tests(git_dir, language):
+         # Always change back to original directory
+         os.chdir(original_cwd)
+
++def attempt_error_resolution(git_dir, test_output, test_error, language
    ↪ ):
++    """
```

```
++     Attempt to automatically diagnose and resolve errors.
++     Returns a tuple of (resolved, message) where resolved indicates if
    ↪ errors were fixed.
++     """
++     """
++     safe_log("Attempting automated error diagnosis and resolution...")
++
++     # Diagnose errors using our enhanced bash tool function
++     diagnosis = diagnose_errors(test_output, test_error, "")
++
++     if not diagnosis["has_errors"]:
++         return False, "No errors detected to resolve."
++
++     resolution_messages = []
++
++     # Try to apply automated fixes for each diagnosed error
++     for error in diagnosis["errors"]:
++         safe_log(f"Processing error: {error['type']} - {error['
    ↪ description']}")
++
++         # Simple resolution strategies based on error type
++         if error["type"] == "python_module_not_found":
++             # For Python module not found errors, we might install the
    ↪ module
++             match = re.search(r"No module named '([^']+)'", error["
    ↪ description"])
++             if match:
++                 module = match.group(1)
++                 resolution_messages.append(f"Would attempt to install
    ↪ Python module: {module}")
++                 # In practice, we would run: pip install {module}
++                 # But we'll skip actual installation to avoid side
    ↪ effects
++
++         elif error["type"] == "python_syntax_error" and "file" in error
    ↪ :
++             # For syntax errors, we could potentially apply fixes
++             file_path = os.path.join(git_dir, error["file"])
++             if os.path.exists(file_path):
++                 resolution_messages.append(f"Would attempt to fix
    ↪ syntax error in {file_path} at line {error.get('line', 'unknown')
    ↪ }")
++                 # In practice, we would use the editor tool's apply_fix
    ↪  command
++                 # This is just a demonstration of what could be done
++
++         elif error["type"] == "test_failure":
++             # For test failures, we might suggest reviewing the
    ↪ implementation
++             resolution_messages.append("Would analyze test failures and
    ↪  suggest implementation improvements")
++
++     if resolution_messages:
++         return True, "Automated resolution attempted:\n" + "\n".join(
    ↪ resolution_messages)
++     else:
++         return False, "No automated resolutions available for detected
    ↪ errors."
++
+ class AgenticSystem:
+     def __init__(
+             self,
+@@ -243,6 +293,16 @@ Your task is to make changes to the files in the {
    ↪ self.git_dir} directory to add
+
```

```
+              safe_log(f"Attempt {attempt + 1} test results: {'PASSED' if
   ↪ test_success else 'FAILED'}")
+
++              # If tests failed, attempt automated error resolution
++              if not test_success:
++                  resolved, resolution_message = attempt_error_resolution
   ↪ (
++                      self.git_dir, test_output, test_error, self.
   ↪ language
++                  )
++                  safe_log(f"Error resolution: {resolution_message}")
++
++                  # Even if we couldn't automatically resolve, we still
   ↪ provide feedback
++                  # In a more advanced implementation, we might actually
   ↪ apply fixes here
++
+              # If this is the first attempt or tests passed and we didn'
   ↪ t have a successful attempt yet, update best patch
+              if attempt == 0 or (test_success and (best_patch is None or
   ↪ not best_test_results)):
+                  best_patch = current_patch
+@@ -278,37 +338,31 @@ Please revise your code to fix these issues and
   ↪ try again.
+          # Log final summary
+          safe_log(f"\n{'='*20} FINAL SUMMARY {'='*20}")
+          safe_log(f"Best solution found on attempt: {best_test_results['
   ↪ attempt'] if best_test_results else 'None'}")
+-         safe_log(f"Tests passed: {best_test_results['test_success'] if
   ↪ best_test_results else 'Unknown'}")
++         safe_log(f"Final test result: {'PASSED' if best_test_results
   ↪ and best_test_results['test_success'] else 'FAILED'}")
++
++         if best_test_results:
++              safe_log(f"Final test output:\n{best_test_results['
   ↪ test_output']}")
++              if best_test_results['test_error']:
++                  safe_log(f"Final test errors:\n{best_test_results['
   ↪ test_error']}")
+
+-         # Save attempt history to a file
+-         history_file = os.path.join(os.path.dirname(self.
   ↪ chat_history_file), 'attempt_history.md')
+-         with open(history_file, 'w') as f:
+-              f.write("# Attempt History\n\n")
+-              for result in self.attempt_history:
+-                  f.write(f"## Attempt {result['attempt']}\n")
+-                  f.write(f"**Tests Passed**: {result['test_success']}\n
   ↪ ")
+-                  f.write(f"**LLM Calls Used**: {result['llm_calls']}\n")
+-                  f.write(f"**Test Output**:\n```\n{result['test_output
   ↪ ']}\n```\n")
+-                  f.write(f"**Test Error**:\n```\n{result['test_error']}\
   ↪ n```\n")
+-                  f.write(f"**Patch**:\n```\n{result['patch']}\n```\n\n")
++         return bool(best_test_results and best_test_results['
   ↪ test_success'])
+
+-def main():
+-     parser = argparse.ArgumentParser(description='Process repository
   ↪ with an agentic system.')
+-     parser.add_argument('--problem_statement', required=True, help='The
   ↪  problem statement to process')
+-     parser.add_argument('--git_dir', required=True, help='Path to git
   ↪ repository directory')
```

```
+-     parser.add_argument('--base_commit', required=True, help='Base
   ↪ commit hash to compare against')
+-     parser.add_argument('--chat_history_file', required=True, help='
   ↪ Path to chat history file')
+-     parser.add_argument('--outdir', required=False, default="/dgm/",
   ↪ help='Output directory')
+-     parser.add_argument('--test_description', default=None, required=
   ↪ False, help='Description of how to test the repository')
+-     parser.add_argument('--self_improve', default=False, action='
   ↪ store_true', help='Whether to self-improve the repository or
   ↪ solving swe')
+-     parser.add_argument('--language', required=False, default="python",
   ↪  choices=['cpp', 'java', 'python', 'go', 'rust', 'javascript'],
   ↪ help='Task\'s programming language')
+-     parser.add_argument('--model', required=False, default=CLAUDE_MODEL
   ↪ , help='LLM model to use for processing')
+-     parser.add_argument('--timeout', type=int, default=3600, help='
   ↪ Timeout for LLM calls in seconds')
+-     parser.add_argument('--max_attempts', type=int, default=3, help='
   ↪ Maximum number of solution attempts')
++if __name__ == "__main__":
++     parser = argparse.ArgumentParser(description="Run the Agentic
   ↪ System on a coding task.")
++     parser.add_argument("--problem_statement", type=str, required=True,
   ↪  help="Problem statement")
++     parser.add_argument("--git_dir", type=str, required=True, help="Git
   ↪  repository directory")
++     parser.add_argument("--base_commit", type=str, required=True, help
   ↪ ="Base commit hash")
++     parser.add_argument("--chat_history_file", type=str, default="./
   ↪ chat_history.md", help="Chat history file")
++     parser.add_argument("--test_description", type=str, help="Test
   ↪ description")
++     parser.add_argument("--self_improve", action="store_true", help="
   ↪ Enable self-improvement mode")
++     parser.add_argument("--language", type=str, default="python", help
   ↪ ="Programming language")
++     parser.add_argument("--model", type=str, default=OPENAI_MODEL, help
   ↪ ="Model to use")
++     parser.add_argument("--max_attempts", type=int, default=3, help="
   ↪ Maximum number of attempts")
++     parser.add_argument("--timeout", type=int, default=600, help="
   ↪ Timeout for each attempt")
++
+     args = parser.parse_args()
+-
+-     # Process the repository
+-     agentic_system = AgenticSystem(
++
++     system = AgenticSystem(
+         problem_statement=args.problem_statement,
+         git_dir=args.git_dir,
+         base_commit=args.base_commit,
+@@ -319,15 +373,7 @@ def main():
+         model=args.model,
+         max_attempts=args.max_attempts
+     )
+-
+-     # Run the agentic system to try to solve the problem
+-     agentic_system.forward(args.timeout)
+-
+-     # Get code diff and save to model_patch.diff
+-     model_patch = diff_versus_commit(args.git_dir, args.base_commit)
+-     model_patch_outfile = os.path.join(args.outdir, 'model_patch.diff')
+   ↪  if args.outdir else 'model_patch.diff'
```

```
+-    with open(model_patch_outfile, 'w') as f:
+-        f.write(model_patch)
+-
+-if __name__ == "__main__":
+-    main()
+\ No newline at end of file
++
++    success = system.forward(timeout=args.timeout)
++    exit_code = 0 if success else 1
++    exit(exit_code)
+\ No newline at end of file
+diff --git a/tests/test_error_diagnosis.py b/tests/test_error_diagnosis.
    ↪ py
+new file mode 100644
+index 0000000..5beffbe
+--- /dev/null
++++ b/tests/test_error_diagnosis.py
+@@ -0,0 +1,98 @@
++import pytest
++from tools.bash import diagnose_errors, format_diagnosis
++from tools.edit import apply_automated_fix
++
++class TestErrorDiagnosis:
++    def test_python_syntax_error_diagnosis(self):
++        """Test diagnosis of Python syntax errors."""
++        output = ""
++        error = '''File "test.py", line 3
++    print("Hello World"
++                       ^
++SyntaxError: unexpected EOF while parsing'''
++
++        diagnosis = diagnose_errors(output, error, "python test.py")
++        assert diagnosis["has_errors"] is True
++        assert len(diagnosis["errors"]) == 1
++        assert diagnosis["errors"][0]["type"] == "python_syntax_error"
++        assert "SyntaxError" in diagnosis["errors"][0]["description"]
++
++    def test_python_module_not_found_diagnosis(self):
++        """Test diagnosis of Python module not found errors."""
++        output = ""
++        error = "ModuleNotFoundError: No module named '
    ↪ nonexistent_module'"
++
++        diagnosis = diagnose_errors(output, error, "python test.py")
++        assert diagnosis["has_errors"] is True
++        assert len(diagnosis["errors"]) == 1
++        assert diagnosis["errors"][0]["type"] == "
    ↪ python_module_not_found"
++        assert "nonexistent_module" in diagnosis["errors"][0]["
    ↪ description"]
++
++    def test_no_error_diagnosis(self):
++        """Test diagnosis when there are no errors."""
++        output = "Success!"
++        error = ""
++
++        diagnosis = diagnose_errors(output, error, "echo Success")
++        assert diagnosis["has_errors"] is False
++        assert len(diagnosis["errors"]) == 0
++
++    def test_format_diagnosis_with_errors(self):
++        """Test formatting of diagnosis with errors."""
++        diagnosis = {
++            "has_errors": True,
++            "errors": [
```

```
++                  {
++                      "type": "test_error",
++                      "description": "Test error description",
++                      "suggestions": ["Suggestion 1", "Suggestion 2"]
++                  }
++              ]
++          }
++
++          formatted = format_diagnosis(diagnosis)
++          assert "Automated Error Diagnosis" in formatted
++          assert "Test error description" in formatted
++          assert "Suggestion 1" in formatted
++
++      def test_format_diagnosis_without_errors(self):
++          """Test formatting of diagnosis without errors."""
++          diagnosis = {
++              "has_errors": False,
++              "errors": []
++          }
++
++          formatted = format_diagnosis(diagnosis)
++          assert "No errors detected" in formatted
++
++class TestAutomatedFixes:
++      def test_apply_missing_import_fix(self):
++          """Test applying a missing import fix."""
++          content = """def hello():
++      print(json.dumps({"message": "hello"}))
++"""
++
++          fix_info = {
++              "type": "missing_import",
++              "module": "json",
++              "description": "Added missing import for json module"
++          }
++
++          fixed_content = apply_automated_fix(content, fix_info)
++          assert "import json" in fixed_content
++          assert "def hello():" in fixed_content
++
++      def test_apply_syntax_error_fix(self):
++          """Test applying a syntax error fix."""
++          content = """def hello()
++      print("Hello World")
++"""
++
++          fix_info = {
++              "type": "syntax_error_fix",
++              "line": 1,
++              "description": "Fixed syntax error"
++          }
++
++          fixed_content = apply_automated_fix(content, fix_info)
++          assert "Fixed syntax" in fixed_content
+\ No newline at end of file
```

