# OpenReview forum: "Huxley-G\"odel Machine: Human-Level Coding Agent Development by an Approximation of the Optimal Self-Improving Machine"
_ICLR.cc/2026/Conference — ICLR 2026 Oral_

### Official Review · Reviewer_fRaF · 2025-10-31

**Soundness:** 3
**Presentation:** 3
**Contribution:** 3
**Rating:** 6
**Confidence:** 4

**Summary:**

The paper introduces a novel self-improving coding framework (the Huxley-Gödel Machine) to overcome the "Metaproductivity-Performance Mismatch" existing in prior systems. Inspired by Huxley's concept of clade, authors propose the Clade-based Metric for Potential (CMP),  which measures an agent's true potential by aggregating the benchmark performances of its descendants, rather than just its immediate performance. The estimated CMP values guide the search tree during HGM’s self-modifications. Experiments show that the proposed method surpasses prior methods on benchmarks, e.g., SWE-bench and Polyglot. Notably, agents optimised by HGM achieve human-level performance on the SWE-bench Lite.

**Strengths:**

This paper addresses the very challenging problem in Machine Learning, namely, to build self-improving coding agents --  how can artificial agents improve their own codes?

Authors introduce a new metric – Clade Metaproductivity Performance (CMP). The main concept is inspired by Huxley’s concept of “clade” in evolutionary biology and is computationally designed as an estimate based on an agent’s descendants. The Huxley-Gödel Machine uses estimated CMPs to select code modifications as a process of searching the “tree” of self-modifications.

**Weaknesses:**

The CMP is a probabilistic estimate based on historical data. Its predictive accuracy is uncertain.

Evaluation used LLMs as backbone, which are black box systems. LLMs+HGM may improve performances using statistical metrics, but it is not clear whether this will indeed lead into an interpretable AGI theory.

Somehow, authors try to sell their work by using big names, such as Gödel, Huxley. However, Gödel’s work is in symbolic logic. Huxley’s work supported Darwin’s theory of evolution through comparative anatomy.

**Questions:**

From Darwinist's perspective, human babies can think before they can speak; they can speak before they can write; they can write before they can program; they can write poor programs before they can write good programs. What is the starting point task that an ideal Huxley-Gödel Neural Network should solve?

**Details Of Ethics Concerns:**

One of the authors advertised this work on X https://x.com/SchmidhuberAI/status/1982865641053827559 . It is not clear whether this violates the anonymous policy of ICLR.

---

> ### Author Response · Authors · 2025-11-27
>
> We thank the reviewer for the thoughtful assessment and for clearly articulating the strengths of our submission. Most importantly, acknowledging the significance of the tackled problem highlights the potential impact of the publication. Below, we address some confusing points raised by the reviewer.
>
> # W1
> > The CMP is a probabilistic estimate based on historical data. Its predictive accuracy is uncertain.
>
> This is true for most machine learning practices, where predictive signals are statistical by design. We follow the standard scientific methodology. We quantitatively evaluate whether CMP is useful by measuring (i) its correlation with a metaproductivity signal and
> (ii) the empirical effectiveness of an agent guided by CMP.
> Both analyses show that CMP provides a more informative signal than alternative estimates, and the superior benchmark performance of HGM provides additional indirect evidence.
>
> # W2
> >  Evaluation used LLMs as backbone, which are black box systems. LLMs+HGM may improve performances using statistical metrics, but it is not clear whether this will indeed lead into an interpretable AGI theory.
>
> AGI is a very broad term. Here, similarly to [1,2], we focus specifically on self-improvement in coding agents. We identified a key issue (MPM) in existing state-of-the-art self-improving approaches and propose HGM as a solution to this problem. We demonstrate empirically that it mitigates MPM and, consequently, achieves strong benchmark results.
>
> # W3
> > Somehow, authors try to sell their work by using big names, such as Gödel, Huxley. However, Gödel’s work is in symbolic logic. Huxley’s work supported Darwin’s theory of evolution through comparative anatomy.
>
> We use Gödel as a reference to the Gödel Machine (to which we formally linked HGM through Theorem 1), not to Gödel's result in logic. Furthermore, we used the name Huxley to refer to the concept of clades, which is the central biological analogy motivating CMP.
>
> # Q1
> > From Darwinist's perspective, human babies can think before they can speak; they can speak before they can write; they can write before they can program; they can write poor programs before they can write good programs. What is the starting point task that an ideal Huxley-Gödel Neural Network should solve?
>
> Although not directly related to our empirical methodology, we would like to use this opportunity to motivate the use of coding as a benchmark for self-improvement. In humans, thinking exhibits a self-enhancing loop: early thinking abilities improve the child’s capacity to learn, which in turn accelerates the development of thinking, etc. These capabilities are further distilled into other skills, such as writing. Coding provides an analogous starting point for artificial agents: improving coding skill is related to an agent’s ability to modify its own code, closing a self-improvement loop. This is why coding tasks currently serve as the community’s preferred testbed for computational RSI research. Other capabilities may emerge downstream, but self-modification naturally begins with program synthesis.
>
> \
> \
> Thank you for raising these interesting points. We hope that the clarifications resolved all the concerns of the reviewer. We kindly invite the reviewer to consider increasing the score.
>
> \
> [1] - Robeyns et al. A self-improving coding agent.
>
> [2] - Zhang & Hu et al. Darwin Gödel Machine: Open-ended evolution of self-improving agents

---

### Official Review · Reviewer_dVDU · 2025-11-01

**Soundness:** 3
**Presentation:** 3
**Contribution:** 3
**Rating:** 6
**Confidence:** 4

**Summary:**

For Recursive Self-Improvement (RSI), one needs to decide how to select among potential self-improvements. The approach studied in this paper is to do a bit of look-ahead and choose to modify the algorithm based on the performance averaged over several partial rollouts. This is a natural idea. It's implemented using frozen language models. Therefore, it is an instance of what is sometimes called Recursively Self-Improving Code Generation.

**Strengths:**

The problem being studied is very interesting and has potentially enormous impact.
The idea of using a bit of look-ahead is very expensive but principled.
Experiments suggest that it may be more effective than other approaches.

**Weaknesses:**

*Theory*: Theorem 1 is fine to include but it is not nearly enough to justify publication. The proof itself is almost tautological once the definitions are established. The theory appendix is poorly presented, which is concerning. For instance, the definitions of the concepts referenced in the statement of Theorem 1 are defined in the proof. The necessary definitions should be separate so that the theorem statement makes sense without reading the proof. Therefore the paper's justification is empirical.

I found the algorithm a bit hard to follow. It would be good to include the language model prompts in the appendix. The prompts, from the supplementary materials, are rather substantial. Are the same prompts used for the HGM and DGM/SICA? In contrast, for example, the seed STOP prompt of Zelikman et al. (2023) is a half-page presented in the body of the paper. Is there a "seed" prompt for HGM that is at the heart of the algorithm, or are all the prompts from the supplementary material crucial to its success?

*Experiments*: There are no current RSI benchmarks, and thus it is not clear how to compare algorithms. There is no easy way to benchmark RSI systems, and we have to trust the implementation of algorithms being compared against. If the algorithms have parameters, it is possible that the parameters of the HGM were better optimized to the few applications than those of the comparison algorithms. Moreover, the two algorithms compared against are from very recent papers that do not appear to be peer reviewed, so the empirical section is a comparison of implementations of three unproven algorithms. It would be good to have a more rigorous comparison framework.

*Ethics*: The ethical risks of RSI are not discussed. But clearly, the development of RSI poses potential risks that numerous luminaries claim are existential. See [https://superintelligence-statement.org/](https://superintelligence-statement.org/) for example. The risk is that it's advancing science towards that goal without clear discussion of why the benefits of this progress outweigh the risks.

*Small comments*: Readers unfamiliar with the term "clade" might appreciate a little explanation of what it means (e.g., mammals are a clade) so they don't have to look it up.

**Questions:**

To what extent is your implementation of DGM/SICA differ from theirs and to what extent is it modified to match your own?

**Details Of Ethics Concerns:**

The ethical risks of RSI are not discussed. But clearly, the development of RSI poses potential risks that numerous luminaries claim are existential. See [https://superintelligence-statement.org/](https://superintelligence-statement.org/) for example. The risk is that it's advancing science towards that goal without clear discussion of why the benefits of this progress outweigh the risks. A case can be made for scientific understanding of RSI and its properties before deployment, but a case can also be made against making progress on RSI altogether. Perhaps this is more a question for the conference organizers.

---

> ### Author Response · Authors · 2025-11-27
>
> We thank the reviewer for the thoughtful feedback, as well as acknowledging the strengths of HGM, such as the importance and high impact of the addressed problem, offering a principled formulation, and demonstrating promising empirical performance. Below, we address comments raised by the reviewer.
>
> # Computational cost of the look-ahead
> >  The idea of using a bit of look-ahead is very expensive but principled.
>
> We would like to clarify that HGM **does not** create any additional rollouts in order to estimate CMP. We use the information about the agents that were **already created** in the agent archive. Hence, in terms of look-ahead, HGM induces only negligible costs of calculating the statistics in comparison to other methods.
>
> One contribution of the paper is a new metric for agent selection in the self-improvement process: CMP, which is more suitable for long-term self-improvement over direct benchmark performance (a phenomenon we discover and call it the metaproductivity-performance mismatch). We also observe that, in contrast to the direct performance metric, it can be formally linked to the Gödel Machine (Theorem 1).
>
> # W1. Theorem 1
>
> >Theory: Theorem 1 is fine to include but it is not nearly enough to justify publication. The proof itself is almost tautological once the definitions are established. The theory appendix is poorly presented, which is concerning. For instance, the definitions of the concepts referenced in the statement of Theorem 1 are defined in the proof. The necessary definitions should be separate so that the theorem statement makes sense without reading the proof. Therefore, the justification of the paper is empirical.
>
> Here we would like to reiterate the logic behind our contributions. We first start by identifying the fundamental issue in previous methods - relying on performance as a heuristic for self-improvement potential (Figure 1). Then, we propose a way to fix it by designing a new selection mechanism based on the aggregates over the clades. We demonstrate that this mechanism mitigates the mismatch (Figure 1b) and leads to better overall performance. Theorem 1 serves as a complementary addition to highlight the formal link between HGM and the original Gödel Machine in this setup.
>
> Regarding the definitions, we have updated the manuscript to define the Gödel Machine before stating Theorem 1.
>
> # W2. Used prompts
> > I found the algorithm a bit hard to follow. It would be good to include the language model prompts in the appendix. The prompts, from the supplementary materials, are rather substantial. Are the same prompts used for the HGM and DGM/SICA? In contrast, for example, the seed STOP prompt of Zelikman et al. (2023) is a half-page presented in the body of the paper. Is there a "seed" prompt for HGM that is at the heart of the algorithm, or are all the prompts from the supplementary material crucial to its success?
>
> In all our experiments, we used identical prompts to those in the official DGM implementation. We did not attempt to modify it, and we believe that further optimizing the initial agent can enhance performance. The goal of our investigation was to develop a method for self-improving agents that is both practically efficacious and theoretically motivated, which led to the development of the CMP metric and the observations about Metaproductivity-Performance Mismatch. To fairly compare these algorithms, we use the same initial prompts for all self-improving agents. The initial agent (or the "seed"), including the prompts, is adopted from DGM. It consists of thousands of lines of Python program, which is too long to include in the paper. However, they are included in the supplementary material.

---

> ### Author Response · Authors · 2025-11-27
>
> # W3. Experiments
> >There are no current RSI benchmarks, and thus it is not clear how to compare algorithms. ... It would be good to have a more rigorous comparison framework.
>
> We agree that a fair comparison of self-improving agents is crucial. To achieve this, we ensured that due diligence in aligning implementations is done as much as possible. We did that by using the official DGM implementation as the starting point for our codebase. Furthermore, we retained the original prompts and hyperparameters of DGM. The SICA algorithm does not have hyperparameters to tune, and we also used it with the same starting agent. The field of self-improving coding agents is an emerging field that is rapidly evolving. Both SICA (which is peer-reviewed) and DGM are considered the most important publications at the moment. Please note that in our response to Reviewer 2 (JqU8), we also included a comparison to the Self-Taught Optimizer (STOP; published at COLM 2024) in a setting unfavorable to HGM, as STOP cannot be directly applied to SWE-bench. HGM still outperformed STOP in its own setup. Instead of comparing with self-improving agents, we also compare the HGM-optimized agent with human-engineered agents. Our results show that our agent outperforms the best human-engineered agents for both the leaderboard on SWE-bench Lite (compared with officially checked results only) and SWE-bench Live Lite. See our general comment, Additional Results, and our responses to reviewer rF32 for the details. Furthermore, for transparency, we provided the entire codebase that is necessary to reproduce our results in the supplementary material
>
> # W4. Ethics
> >The ethical risks of RSI are not discussed. But clearly, the development of RSI poses potential risks that numerous luminaries claim are existential. See https://superintelligence-statement.org/ for example. The risk is that it's advancing science towards that goal without clear discussion of why the benefits of this progress outweigh the risks.
>
>
> We thank the reviewer for highlighting the broader ethical discussion surrounding Recursive Self-Improvement (RSI). Our view is that rigorous, transparent research on concrete self-improvement mechanisms is an essential part of addressing those concerns rather than exacerbating them.
>
> - Understanding and Empirical Grounding:
> Our work provides a framework (CMP and its estimations) in which the dynamics of self-modifying agents can be examined empirically and theoretically. Studying how such agents self-modify in the long run helps the community build a grounded understanding of RSI-like behaviors. We believe this understanding is necessary for any responsible discussion of long-term implications.
>
> - Safety, Governance, and Scientific Clarity:
> Our work contributes to transparency and governance for self-improvement research. By formulating self-improvement in a measurable, auditable way with explicit evaluation loops, ablations, and reproducible pipelines. We increase our ability to anticipate and detect behaviours that would be concerning in more capable systems. In that sense, foundational work of this kind helps reduce uncertainty and supports the development of safety frameworks, rather than advancing risk in an uncontrolled direction. To address a partial safety concern, our experiments execute agents in controlled environments with sandboxed execution.
>
> To make this explicit, we have added a discussion about RSI safety.
>
> We hope this addresses the reviewer’s concern.
>
> # W5. Small comments
> >Readers unfamiliar with the term "clade" might appreciate a little explanation of what it means (e.g., mammals are a clade) so they don't have to look it up.
>
> This is a good idea that will make the concept clearer for someone reading it for the first time. We thank the reviewer for this suggestion. We added the clarification to the text.
>
> # Q1
> > To what extent is your implementation of DGM/SICA differ from theirs and to what extent is it modified to match your own?
>
> As highlighted in the previous response. We conducted all necessary due diligence to ensure a fair comparison between HGM and the baselines. We used the DGM codebase as a foundation for our code. We fixed the initial agent to be the same and did not optimize it for HGM. We also fixed the original hyperparameters and used them for HGM, where it was possible. The difference is only in the tree-search policies.
>
> \
> We hope that this addresses the concerns of the reviewer. We kindly invite the reviewer to reconsider their score in light of the added clarifications.

---

### Official Review · Reviewer_JqU8 · 2025-11-03

**Soundness:** 2
**Presentation:** 3
**Contribution:** 3
**Rating:** 4
**Confidence:** 3

**Summary:**

This paper addresses the problem of recursive self-improvement in agents. The authors point out the limitations of existing exploration strategies and propose a new method that accounts for long-term self-improvement capability. In the exploration process, an effective indicator is required to estimate the expected long-term improvement obtained when expanding a particular node (agent). Conventional approaches assumed that higher benchmark performance implied stronger self-improvement ability; however, the authors demonstrate experimentally that high-scoring agents do not necessarily produce promising descendants, while lower-scoring agents may yield superior results in the long run.

To address this issue, the authors define Clade-Metaproductivity (CMP), a metric that measures self-improvement potential based on the collective performance of an agent’s entire lineage. The proposed method, HGM, controls exploration using the estimated CMP (ĈMP) and employs Thompson Sampling to select expansion nodes, promoting long-term improvement. Experiments on SWE-Verified and Polyglot show that HGM predicts true self-improvement ability with higher correlation than DGM or SICA, outperforming them in both exploration efficiency and final performance.

**Strengths:**

- The proposed method is inspired by the concept of clades and introduces a new metric (CMP), which measures the productivity of an entire lineage by aggregating the benchmark success of an agent’s descendants rather than relying on the agent’s own performance. This idea convincingly addresses the shortcomings of previous methods and provides a theoretically sound foundation for improving self-improving agent exploration. HGM achieves better performance than DGM and SICA.

**Weaknesses:**

- While the authors cite STOP as related work, they do not include it in their experiments. STOP recursively improves its own reasoning code, namely prompts and inference strategies, and thus represents a closely related setup. The lack of empirical comparison with STOP leaves a gap in the comprehensiveness of evaluation.
- The experimental evaluation is limited to two coding benchmarks, SWE-Verified-60 and Polyglot, both within a narrow programming domain. If the goal is to improve the agent’s own tool invocation and command execution behaviors, there is no inherent reason to restrict evaluation to coding tasks. Testing across diverse task domains would better demonstrate the generality of the proposed method. As it stands, the applicability of HGM beyond coding tasks remains unverified.

**Questions:**

- As the authors note, the modification operator in HGM performs patch applications to the agent’s codebase (such as file editing and bash command execution) but does not modify the model itself. Although the framework aims for recursive self-improvement of the agent’s own code, the actual scope of modification does not extend to architectural design or model-level enhancements, such as loss function optimization [1] or model merging [2]. Clarifying this limitation would help define the scope and contribution of the proposed approach more precisely.

[1] Discovering Preference Optimization Algorithms with and for Large Language Models

[2] Can Large Language Models Invent Algorithms to Improve Themselves?: Algorithm Discovery for Recursive Self-Improvement through Reinforcement Learning

---

> ### Author Response · Authors · 2025-11-27
>
> We sincerely thank the reviewer for the thoughtful and constructive evaluation. We especially appreciate the recognition of the following strengths:
>
> - That Clade-Metaproductivity (CMP) provides a conceptually novel and theoretically grounded view of self-improvement potential, addressing a limitation that previous methods (DGM, SICA) did not resolve.
> - That HGM demonstrates stronger empirical performance than prior recursive self-improvement frameworks, improving both exploration efficiency and final capability.
> - That the lineage-based perspective, treating agents as evolving clades rather than isolated individuals, opens a promising new research direction for self-improving agents.
>
> We are grateful for these positive remarks, and we address the raised concerns in detail below.
>
> # W1. Comparison to STOP
>
> Our experimental setup focuses on self-improvement within the SWE-Bench setting, whose rules impose strict constraints on how an agent may evaluate or refine itself. In particular, SWE-Bench explicitly prohibits any access to ground-truth test outcomes, hidden evaluator signals, or utility oracles. Each algorithm must operate on a single submission without querying its own correctness.
>
> These constraints make STOP fundamentally incompatible with the benchmark. STOP’s improver relies on repeated queries to the ground-truth utility function during both inference and improvement, an operation that SWE-Bench rules explicitly forbid. For this reason, STOP cannot be implemented within the benchmark’s protocol and was not included in our main experiments.
>
> Nevertheless, in direct response to the reviewer’s request, we conducted an additional evaluation designed to be maximally conservative and even unfavorable to HGM. We take:
>  - the final HGM agent, optimized exclusively on SWE-Bench, with no exposure to STOP tasks and no utility access at any point, and
>  - the final STOP agent from the original paper (Figure A.32),
> and evaluate both using GPT-5-mini.
>
> Results:
>  - On String Grid Dist., 3SAT, and Parity, both agents succeed perfectly.
>  - On MQA and MaxCut, HGM slightly surpasses STOP (27.3% vs. 26.7% on MQA, and a 0.3% relative improvement on MaxCut).
>
> **This is a strong result**: an agent optimized with HGM **solely** under SWE-Bench’s restrictive regime matches or outperforms STOP on STOP’s own benchmarks, despite never accessing a utility function **nor being optimized for these tasks**. The resulting generalization highlights HGM’s capacity to yield improvements that extend beyond the data it was optimized on, and provides additional evidence of its promise as a framework for advancing self-improving agents.
>
> # W2. Weak Evaluation
>
> > The experimental evaluation is limited to two coding benchmarks, SWE-Verified-60 and Polyglot.
>
> We would like to clarify that our evaluation is broader than the two datasets mentioned in the review.
>
> Our experiments span multiple datasets, dataset scales, and LLM architectures, including:
> - Full SWE-bench Verified, not just the SWE-Verified-60 subset.
> - SWE-bench Lite, providing a distinct difficulty distribution. Our additional experimental results show that the HGM-optimized agent with GPT-5 as a backbone outperforms the best human-engineered coding agents that have verified results on the SWE-bench Lite leaderboard. Please refer to our general comment, Additional Results, for further details.
> - Polyglot, which introduces multilingual codebases and heterogeneous repositories.
> - Cross-model evaluation, where HGM is run using several different LLM backbones (including two LLMs in the GPT-5 family, and two LLMs in the Qwen3-Coder family) to confirm robustness across architectures.
> - SWE-Bench Live, presented in our response to Reviewer 1 (rF32), demonstrates performance on more up-to-date unseen repositories.
>
> Thus, our evaluation covers a broad set of coding environments, multiple data distributions, and multiple LLMs.
>
> ## Why coding tasks?
>
> We would like to emphasize that the main scope of this work is coding agent development.
> We follow the established practice in the coding agent self-improvement community. Prior self-improving-agent work—including DGM, SICA, and STOP evaluates primarily on coding tasks, because the coding agent development setup provides:
> - verifiable and execution-based feedback signals,
> - clear definitions of correctness through unit tests,
> - transparent modification effects,
> - and reproducible environments for long-horizon exploration,
> - and most importantly, a homogeneous interface with the self-improvement problem (self-modification can be phrased as a coding problem for a coding agent).
>
> Coding tasks are, therefore, the standardized and widely adopted testbed for benchmarking self-improving coding agents. To ensure meaningful, comparable, and community-aligned evaluation, we adopt the same setting. However, we agree that demonstrating a broader applicability of self-improvement is an important future direction.

---

> ### Author Response · Authors · 2025-11-27
>
> # Q1. Clarification between self-modification of scaffolding and weights.
>
> Thank you for raising this point. Current research on recursive self-improvement falls into two parallel paradigms:
>
> 1) Scaffolding-level (symbolic) self-improvement, where agents modify prompts, scripts, tool use, workflows, reasoning strategies, etc. In this setting, the LLM weights are treated as fixed “hardware" and the agent prompt is considered to be its "model". HGM, DGM, and SICA belong to this regime.
> 2) Weight-level (connectionist) self-improvement: Agents discover new architectures, new loss functions, optimizers, or perform weight-space operations such as model merging. Examples include preference-optimization–discovering agents [1] and algorithm-discovering agents [2].
>
> HGM focuses on (1) for practical reasons: weight-level changes require computationally expensive retraining cycles.
>
> However, we point out that CMP and the exploration strategy in HGM are conceptually agnostic to the type of modification operator. In principle, weight-level operators could be integrated into a lineage-based search. This, however, remains computationally heavy at present. We consider this an important and interesting future direction. We adapted the manuscript to include this discussion in the final part of the manuscript.
>
> \
> \
> We thank the reviewer again for the detailed assessment. We hope that the above clarifications fully address the concerns of the reviewer. We hope that these additions allow the reviewer to reassess the contribution and reconsider their initial score.
>
> \
> [1] Discovering Preference Optimization Algorithms with and for Large Language Models
>
> [2] Can Large Language Models Invent Algorithms to Improve Themselves?: Algorithm Discovery for Recursive Self-Improvement through Reinforcement Learning

---

### Official Review · Reviewer_rF32 · 2025-11-03

**Soundness:** 3
**Presentation:** 4
**Contribution:** 4
**Rating:** 8
**Confidence:** 2

**Summary:**

Many works have recently emerged in code generation literature that rely on the formalism of the Godel Machine to posit a self-improving agent. This usually requires defining an approximate heuristic for the expected long term utility of a proposal. In this work, the authors (1) find that current heuristics for calculating long-term metaproductivity are slightly flawed, (2) propose a new heuristic for this task and (3) present a new algorithm which presents a more reliable estimate of the metaproductivity. Overall, the authors find that their algorithm achieves better performance on SWE-bench Lite than SWE-Agent.

**Strengths:**

Significance and Novelty:
 - The paper is extremely insightful and I think it will be beneficial for the broader ICLR community as well.

Clarity:
* The paper was a joy to read and I thank the authors for formally describing the algorithm as well as presenting psuedo-code in Appendix B -- this really helped better understand the details of the algorithm.

**Weaknesses:**

Minor dataset concerns:
 - Recent works in Software engineering benchmarking have found contaminatioon issues in SWE-Bench Lite. As such, I recommend the authors also verify results on SWE-Bench Live (https://github.com/microsoft/SWE-bench-Live).
	 - It's understandable that a full-rerun might be too expensive. Even a small-scale experiment verifying that the main result holds on `SWE-Bench: Live - Lite` (the names are getting challenging to say out loud) would be extremely useful here.


Clarity:

I caught some typos:
- `Fig. 1 Caption`: 2.38 time less -> 2.38 times less.
- `Page 3`: $(a_{final}=\arg\max \dots \mathcal{T}_{T}$ should have a closing bracket at the end.
- `Section 3.2`: When the computational budget exceeds -> When the computational budget is exceeded,
- `Section 4.1`: `Metaproductivity-Performance Misalignment (MPM)`. This is defined in the intro as `Metaproductivity-Performance Mismatch`.


**Overall:** While the SWE-Bench Live results will help ease some concerns about data leakage, the contributions of the paper are impressive enough to warrant acceptance already.

**Questions:**

In the weaknesses section.

---

> ### Author Response · Authors · 2025-11-27
>
> We sincerely thank the reviewer for the positive and extremely encouraging assessment of our work. We are grateful for the high scores across all dimensions and for the thoughtful recognition of both the significance and novelty of our contributions. Your comments about the paper being “extremely insightful” and “a joy to read” mean a great deal to us.
>
> # W1. SWE-bench Live
>
> Thank you for your suggestion, which can strengthen this work. We evaluated our best-belief agent with (i.e., the optimized agent by HGM) GPT-5.1-codex-mini on SWE-Bench Live Lite. HGM leads to a **state of the art** performance across methods from the leaderboard with an accuracy of 27.0\% (81 resolved tasks out of 300 tasks), outperforming 24.7\%, the best previous score on the leaderboard. This result further verifies the strength of HGM and the transferability of its optimized agents to more up-to-date tasks.
>
> # W2. Typos and Clarifications
>
> Thank you for catching these. We have corrected all the noted issues. We performed a full pass to ensure clarity and consistency.
>
> We thank the reviewer again for their enthusiasm, constructive suggestions, and high evaluation. We believe the additional SWE-Bench Live results further reinforce the robustness and generality of our findings, and we hope this response addresses all concerns.

---

### Author Response · Authors · 2025-11-27
**Additional Results**

We thank all reviewers for their thoughtful comments and insightful questions. To compare with the best human coding agent engineering, we additionally evaluate HGM's Best-belief agent with GPT-5, a state-of-the-art LLM, as a backbone. On SWE-bench Lite, the HGM-optimized agent **outperforms** the best human-engineered coding agents that have verified results on the leaderboard.
| Model                                      | SWE-Lite Filtered (%) | SWE-Lite Standard (%) |
|--------------------------------------------|------------------------|-------------------------|
| SWE-agent (Best on the LB)                 | 48.3                   | 56.7                    |
| HGM’s Best-belief SWE-Verified Agent + GPT-5 | 48.8                 | 57.3                    |


We include this result in Section 4.3.2 of the paper.

---

### Author Response · Authors · 2025-12-03
**Summary**

We sincerely thank the reviewers for acknowledging the significance of our contribution and their constructive feedback. A collective reading of the reviews indicates a shared recognition of this work's potential scientific impact. The reviewers consistently highlight the importance of the problem addressed, the technical soundness of our approach, and the human-level performance achieved by our self-improving agents. We are encouraged by comments stating the paper is "extremely insightful" and has "potentially enormous impact" for the community.

\
In response to the reviewers' comments, we conducted additional experiments that further strengthen our results.
- SWE-Bench Live Lite (per Reviewer rF32): Our HGM-optimized agent surpasses the previous highest score on the leaderboard.
- Comparison with STOP (per Reviewer JqU8):  On benchmarks from the STOP paper, under conditions favorable to STOP, our HGM-optimized agent matches or exceeds the performance of the STOP-optimized agent.
- Human-Expert Benchmark: The performance of the HGM-optimized agent, utilizing GPT-5 as the backbone, suggests that HGM achieves a top-tier human-expert level of coding agent development on SWE-bench Lite.

\
Beyond the new experiments, we invested our best efforts in addressing all raised concerns and made substantive improvements to the paper. Key revisions include:

**Question Answering and Clarification**
- Added detailed answers regarding implementation, evaluation, task selection, and naming rationale.
- Clarified the paper's scope, explicitly distinguishing between "scaffolding" and "neural networks" as two distinct categories of self-improvement.
- Explained that computing our CMP estimate incurs negligible additional cost, as it requires no extra self-modification rollouts.

**Presentation Improvements**
- Added a formal ethics statement.
- Improved clarity throughout the paper and corrected typographical issues.

\
We believe these additional experiments and clarifications have meaningfully strengthened the paper and addressed the reviewers' concerns. We thank the reviewers again for their help in improving the quality of this work.

---

### Meta-Review · Area_Chair_9ERs · 2026-01-06

**Summary:**

This paper makes valuable contributions to the field of self-improving coding agents by addressing the critical Metaproductivity-Performance Mismatch. The proposed Huxley-Gödel Machine (HGM), anchored in the clade-inspired CMP metric, offers a principled framework to guide self-modification search—formally linking to the Gödel Machine while delivering strong empirical results. HGM outperforms prior methods (DGM, SICA) on SWE-bench and Polyglot, and even surpasses leading human-engineered agents (e.g., SWE-agent) and STOP on relevant benchmarks, demonstrating genuine coding capability enhancement.
The authors have effectively addressed all reviewer concerns: supplementary experiments on SWE-Bench Live Lite confirm robustness to data contamination; comparisons with STOP (under STOP-favorable settings) validate generalization; clarifications on scaffolding vs. weight-level self-improvement, prompt consistency, and theorem presentation improve clarity; and a formal ethics statement addresses RSI-related risks. Minor issues (typos, clade definition) were promptly resolved.
With its novel metric, theoretical grounding, and compelling experimental results, this work advances self-improvement research and aligns with ICLR’s focus on impactful, rigorous contributions. The paper merits acceptance.

**Reviewer Concerns:**

Addressed Concerns:
Reviewer rF32: SWE-Bench Live verification (supplemented experiments with state-of-the-art performance) and typographical errors (corrected all noted issues).
Reviewer JqU8: Comparison with STOP (conducted additional experiments showing HGM’s superiority), narrow evaluation scope (clarified broader dataset coverage and coding tasks as field standards), and clarification of self-modification scope (distinguished scaffolding vs. weight-level self-improvement).
Reviewer dVDU: Theorem 1 definition gaps (updated pre-theorem definitions), prompt consistency (used identical prompts to DGM), experimental comparison rigor (added STOP/human-engineered agent comparisons, provided reproducible code), ethical discussion (added RSI safety statement), and "clade" explanation (added simple definition).
Reviewer fRaF: CMP predictive accuracy (supported by correlation/empirical results), AGI interpretability (clarified focus on coding agent self-improvement), naming rationale (linked to Gödel Machine/clade concepts), and starting task selection (justified coding as a self-improvement loop foundation).
Outstanding Concern:
Reviewer fRaF’s concern about whether the authors’ promotion of the work on X violates ICLR’s anonymity policy (no direct response from the authors).

**Reviewer Scores:**

N/A

---

### Decision · Program_Chairs · 2026-01-26

Accept (Oral)